# Falcon: Fast Proximal Linearization of Normalized Cuts for Unsupervised Image Segmentation

**Xiao Zhang**
University of Pennsylvania

**Xiangyu Han**
University of Pennsylvania

**Xiwen Lai**
University of Pennsylvania

**Yao Sun**
Hong Kong Polytechnic University

**Pei Zhang**
Wuhan University

**Xia Liu**
University of Wollongong

**Konrad Kording**
University of Pennsylvania

## Abstract

Current zero-shot unsupervised segmentation methods based on normalized cuts (NCut) face three key limitations. First, they rely on recursive bipartitions with repeated eigen-decompositions, making them prohibitively expensive at scale. Second, each split requires spectral relaxation followed by rounding, introducing layers of approximation where the final partition may diverge from the true NCut objective. Third, recursive bipartitioning offers no principled assurance of producing a stable $K$-way segmentation, and existing heuristics lack convergence guarantees. We propose **Falcon**, a proximal-gradient solver that directly optimizes the discrete $K$-way NCut objective without spectral relaxation. We prove linear convergence under the *Kurdyka–Łojasiewicz* (KL) property. Falcon computes closed-form gradient scores weighted by cluster volumes and performs row-wise one-hot proximal updates stabilized by inertia. A monotone backtracking scheme adaptively tunes the proximal parameter, ensuring non-decreasing NCut values. This design preserves discrete feasibility, removes repeated eigen-decomposition, and guarantees convergence. Across six benchmarks, Falcon outperforms the strongest official baseline (DiffCut) by wide margins, e.g., +13.2 mIoU on VOC, +27.7 on COCO-Object, and +3.1 on Cityscapes, while remaining competitive on Pascal Context. It also runs up to an order of magnitude faster than recursive NCut and scales more favorably in memory at high resolution, making it practical for larger token grids. By pairing pretrained foundation models with a principled NCut solver, Falcon sets a new state of the art across six benchmarks and achieves the best performance on 17 of 18 benchmark–encoder pairs, underscoring both its robustness and its generality in bridging the gap between unsupervised and supervised segmentation. [1]

## 1 Introduction

Semantic segmentation partitions an image into regions whose pixels share the same semantic meaning, such as belonging to the same object or stuff category. As a fundamental computer vision task, it underpins a wide range of downstream applications, including image editing, medical imaging, scene understanding, and autonomous driving Wang et al. (2023a); Zhu et al. (2016); Esser et al. (2023); Zhou et al. (2019b); Li et al. (2022b); Han et al. (2024). Over the past decade, semantic segmentation has advanced rapidly, driven first by Fully Convolutional Networks (FCNs) Long et al. (2015) and later by architectures such as SegNet Hu et al. (2018), U-Net Wang & Yang (2021), DeepLab Chen et al. (2017), and PSPNet Zhao et al. (2017), which improved multi-scale reasoning and feature fusion. More recently, Transformer-based models Dosovitskiy et al. (2020); Oquab et al. (2023); Caron et al. (2021); Xie et al. (2021); Cheng et al. (2022) and foundation segmentation models such as SAM Ravi et al. (2024) have substantially strengthened segmentation quality and flexibility.

---

[1]Code is available at `https://github.com/ZhangXLaurence/Falcon-Seg`.

Despite this progress, most high-performing segmentation systems rely on dense pixel-level annotations, which are expensive and labor-intensive to obtain. This has motivated growing interest in unsupervised and zero-shot segmentation, where the goal is to discover semantically meaningful regions without task-specific mask supervision, often for categories not seen during training Tian et al. (2024); Couairon et al. (2025); Sick et al. (2024). Existing approaches roughly fall into two groups. One line relies on representation learning and clustering objectives, such as STEGO Hamilton et al. (2022), IIC Ji et al. (2019), and PiCIE Cho et al. (2021), which learn or refine patch-level groupings through self-supervision. Another line combines pretrained visual features with classical grouping principles such as graph partitioning and normalized cuts (NCut), yielding strong training-free segmentation pipelines Wang et al. (2023b;a); Couairon et al. (2025).

In this paper, we focus on the *training-free* zero-shot setting: a frozen pretrained encoder is used only at inference time, without mask annotations, task-specific finetuning, or adaptive self-training. This setting is particularly attractive because it separates representation learning from mask generation and enables segmentation to be performed directly from general-purpose foundation features. Among training-free methods, graph-based segmentation has recently shown strong performance. TokenCut Wang et al. (2023b) and MaskCut Wang et al. (2023a) recursively apply normalized cut to token affinities extracted from self-supervised Vision Transformers, while DiffCut Couairon et al. (2025) further improves connectivity by incorporating diffusion priors. These methods demonstrate that strong pretrained features combined with graph partitioning can already produce competitive unsupervised masks.

However, current NCut-based training-free pipelines still face three key limitations. *First*, they typically rely on recursive bipartitions, where each split requires a new eigen-decomposition; this makes inference increasingly expensive as the token graph grows. *Second*, they optimize a spectrally relaxed problem and then round the solution back to discrete labels, introducing layers of approximation such that the final partition may deviate from the original discrete $K$-way NCut objective. *Third*, recursive bipartitioning provides no principled assurance of producing a stable $K$-way segmentation, and the resulting heuristics generally come without convergence guarantees. Together, these issues create a gap between the elegant combinatorial objective of NCut and the practical procedures used in modern zero-shot segmentation systems.

To address this gap, we propose **Falcon**, a fast proximal linearization method for directly optimizing the discrete $K$-way NCut objective. Rather than recursively splitting the graph through spectral bipartitions, Falcon maintains a global $K$-way one-hot assignment throughout optimization and updates it via closed-form row-wise proximal steps derived from the exact gradient of the objective. A monotone backtracking scheme adaptively adjusts the proximal parameter to ensure non-decreasing objective values, while the discrete one-hot structure is preserved at every iteration. In this way, Falcon removes repeated recursive eigen-decompositions from the solver itself, avoids relax-and-round mismatch, and provides a principled optimization framework with convergence guarantees under the *Kurdyka–Łojasiewicz* (KL) property.

Beyond its theoretical appeal, Falcon is also highly practical. Because it is fully vectorized and avoids recursive NCut passes, it substantially reduces runtime compared with prior graph-cut baselines. Moreover, while dense token affinities remain quadratic in the number of tokens, Falcon scales more favorably in memory than prior NCut-based methods at high resolution, making it more practical for larger token grids.

Our contributions are summarized as follows:

**Method.** We introduce Falcon, a proximal-gradient solver for the discrete $K$-way NCut problem that preserves one-hot feasibility throughout optimization. Each iteration computes closed-form gradient scores weighted by cluster volumes and performs row-wise one-hot proximal updates stabilized by inertia. This design directly optimizes the original discrete objective without spectral relaxation or recursive bipartitioning.

**Convergence.** We show that Falcon admits a monotone backtracking scheme with non-decreasing objective values and prove convergence under the *Kurdyka–Łojasiewicz* (KL) framework. In particular, the iterates have finite length and converge to a critical point; when the KL exponent satisfies $\theta \leq \frac{1}{2}$, the local convergence rate is at least linear.

**Performance.** Across six benchmarks and three pretrained encoders, Falcon achieves state-of-the-art unsupervised segmentation performance, obtaining the best results on **17 of 18** benchmark–encoder pairs. It also runs up to an order of magnitude faster than recursive NCut baselines and

exhibits more favorable memory scaling at high resolution, demonstrating both robustness and practicality in training-free segmentation.

## 2 RELATED WORKS

**Vision Foundation Models.** Vision foundation models typically leverage unlabeled data to learn robust and generalizable representations. Early contrastive methods like MoCo He et al. (2020) and BYOL Grill et al. (2020) laid the groundwork for advanced frameworks such as SwAV Goyal et al. (2021), DINO Caron et al. (2021); Oquab et al. (2023); Darcet et al. (2023); Siméoni et al. (2025), and iBOT Zhou et al. (2021), while masked autoencoders He et al. (2022) have further refined reconstruction-based pre-training. Beyond purely visual approaches, multimodal pre-training has surged in prominence, with models like CLIP Radford et al. (2021), BLIP Li et al. (2022a), and SigLIP Zhai et al. (2023); Tschannen et al. (2025) aligning high-level image features to text. In parallel, diffusion-based methods such as Stable Diffusion Rombach et al. (2021); Gupta et al. (2024) extend these capabilities by learning rich generative representations, enabling tasks ranging from zero-shot classification to semantic correspondence. Collectively, these developments highlight the efficacy of foundation models in scaling to large, diverse datasets and adapting readily to downstream tasks.

**Semantic Segmentation.** Semantic segmentation partitions an image into semantically coherent regions by labeling each pixel, enabling the understanding of the structured scene. It is broadly categorized into supervised and unsupervised methods. Supervised segmentation, extensively studied and achieving high accuracy Namekata et al. (2024); Wang et al. (2021); Cheng et al. (2021); Ravi et al. (2024), relies on large-scale annotated datasets. Recent work has explored text-based supervision to mitigate the need for dense annotations Ranasinghe et al. (2023); Xu et al. (2022); Cha et al. (2023); Ren et al. (2023). In contrast, unsupervised methods often require dataset-specific training to achieve competitive performance Liang et al. (2023); Feng et al. (2023); Cho et al. (2021); Li et al. (2023), and zero-shot segmentation for unseen categories remains challenging. DiffSeg Tian et al. (2024) leverages self-attention maps from a pre-trained diffusion model, applying KL-divergence-based iterative merging for segmentation. DiffCut Couairon et al. (2025) improves upon this by extracting richer encoder features from the self-attention block of a Transformer and incorporating a recursive N-Cut Shi & Malik (2000a) algorithm.

**Graph-based Image Segmentation.** Early approaches, such as Normalized Cut (N-Cut) Shi & Malik (2000a), optimized the segmentation problem through spectral graph theory, formalizing spectral clustering theory based on the N-Cut objective; yet, its computational limitations persisted. To improve efficiency and adaptability, F & P (2004) proposed an adaptive merging strategy that utilizes both intra-region and inter-region criteria, while Grady (2006) introduced a probabilistic random walk framework that leverages adjacent pixel relationships to handle complex textures and weak boundaries. Recent deep learning-integrated approaches, such as TokenCut Wang et al. (2022), AutoSC Fan et al. (2022), and DiffCut Couairon et al. (2025), compute token-level similarities via self-supervised Transformer features but remain constrained by N-Cut's recursive bisection strategy and hard segmentation constraints.

**Proximal Gradient Methods.** Proximal gradient algorithms handle composite objectives that pair a smooth data term with a possibly nonsmooth regularizer frequently seen in learning (e.g., sparsity, total variation, simple constraints). A gradient step on the smooth part followed by a proximal step on the regularizer yields a simple, scalable routine with standard convergence guarantees and practical backtracking/line–search implementations Parikh & Boyd (2014). The accelerated variant FISTA achieves the optimal $\mathcal{O}(1/k^2)$ rate for convex problems, and monotone variants stabilize the objective along the iterates Beck & Teboulle (2009); Chambolle & Dossal (2015). For nonconvex formulations common in modern models, convergence to a critical point can be justified under the Kurdyka–Łojasiewicz framework and via blockwise/prox–linear updates Attouch et al. (2013); Bolte et al. (2014). Inertial extensions further improve empirical speed while retaining convergence guarantees under mild conditions Ochs et al. (2014).

## 3 METHODOLOGY

### 3.1 SEGMENTATION AS NCUT ON TOKENS

Graph-based segmentation casts image segmentation as partitioning a weighted graph. Following normalized-cut pipelines Wang et al. (2022; 2023a); Couairon et al. (2025), we build an undirected

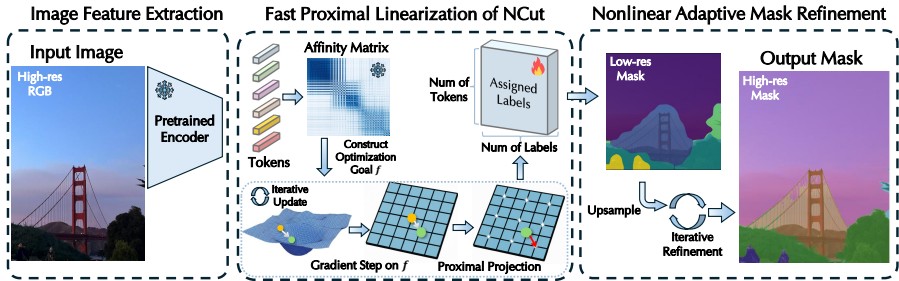

Figure 1: **Overview of Falcon.** (1) Image feature extraction: tokens are extracted from the input image. (2) Fast proximal linearization of NCut: an affinity matrix is constructed between tokens, and the discrete $K$-way assignment is iteratively updated by row-wise one-hot proximal steps. (3) Optional mask densification and refinement: the coarse token-level mask is first densified on an intermediate grid and can then be paired with a lightweight pixel-level refiner such as PAMR or NAMR.

graph $G = (V, E)$ with $N$ nodes, one per $d$-dimensional token from a Vision Transformer:

$$V = \{\boldsymbol{f}_1, \ldots, \boldsymbol{f}_N\}, \quad \boldsymbol{f}_i \in \mathbb{R}^d, \qquad \boldsymbol{F} = [\boldsymbol{f}_1 \ \cdots \ \boldsymbol{f}_N]^\top \in \mathbb{R}^{N \times d}. \tag{1}$$

We use a dense affinity graph (without self-loops) and form a nonnegative, symmetric similarity by row-normalizing features to unit $\ell_2$ norm (cosine similarity), rescaling to $[0, 1]$, and applying elementwise power sharpening:

$$\widehat{\boldsymbol{F}} = \text{row\_norm}(\boldsymbol{F}), \quad \boldsymbol{A} = \widehat{\boldsymbol{F}}\widehat{\boldsymbol{F}}^\top, \quad \boldsymbol{W} = \phi(\boldsymbol{A})^{\odot \alpha}, \ \alpha \geq 1, \qquad \text{diag}(\boldsymbol{W}) \leftarrow \boldsymbol{0}, \tag{2}$$

where $\phi$ maps similarities to $[0, 1]$ (e.g., $\phi(s) = \max(s, 0)$ or min–max scaling per image). We then define

$$\boldsymbol{d} = \boldsymbol{W}\boldsymbol{1}, \qquad \boldsymbol{D} = \text{diag}(\boldsymbol{d}). \tag{3}$$

Let $\{P_1, \ldots, P_K\}$ be a partition and let $\boldsymbol{x}_k \in \{0, 1\}^N$ denote the indicator vector of partition $P_k$. Stacking these indicators gives a row-wise one-hot assignment matrix $\boldsymbol{X} = [\boldsymbol{x}_1, \ldots, \boldsymbol{x}_K] \in \{0, 1\}^{N \times K}$ satisfying $\boldsymbol{X}\boldsymbol{1} = \boldsymbol{1}$, i.e., each token is assigned to exactly one cluster. The normalized cut objective Shi & Malik (2000b) is

$$\text{Ncut}(P_1, \ldots, P_K) = \sum_{k=1}^{K} \frac{\text{cut}(P_k, \bar{P}_k)}{\text{vol}(P_k)}, \quad \text{cut}(P_k, \bar{P}_k) = \sum_{i \in P_k, \ j \notin P_k} W_{ij}, \quad \text{vol}(P_k) = \sum_{i \in P_k} d_i. \tag{4}$$

Equivalently, minimizing NCut is the same as maximizing the normalized association (i.e., a sum of $K$ generalized Rayleigh quotients)

$$f(\boldsymbol{X}) = \sum_{k=1}^{K} \frac{\boldsymbol{x}_k^\top \boldsymbol{W} \boldsymbol{x}_k}{\boldsymbol{x}_k^\top \boldsymbol{D} \boldsymbol{x}_k}, \qquad \text{Ncut}(\boldsymbol{X}) = K - f(\boldsymbol{X}). \tag{5}$$

The problem is discrete and nonconvex; finding the exact optimum is NP-hard, which motivates principled relaxations and iterative schemes.

## 3.2 FALCON: FAST PROXIMAL LINEARIZATION OF NCUT

Classical NCut pipelines typically optimize a relaxed continuous problem, extract one or a few eigenvectors, and then convert the result back to discrete labels through recursive bipartitioning and rounding. Falcon instead operates directly on a global discrete $K$-way assignment. At each iteration, it evaluates how favorable it is to assign each token to each cluster under the exact gradient of the normalized association objective, and then projects each row back to a valid one-hot label. A proximal term acts as an inertia mechanism, discouraging unnecessary label flips and stabilizing the discrete updates, while a monotone backtracking rule automatically adjusts its strength to ensure non-decreasing objective values. In this way, Falcon performs principled local search on the original discrete NCut objective, rather than relying on spectral relaxation and recursive splitting.

Rather than relaxing the row-wise one-hot constraints, we model discreteness through the extended-valued composite objective

$$\min_{\boldsymbol{X} \in \mathbb{R}^{N \times K}} \Phi(\boldsymbol{X}) = h(\boldsymbol{X}) + g(\boldsymbol{X}), \qquad h(\boldsymbol{X}) = -f(\boldsymbol{X}), \qquad g(\boldsymbol{X}) = \iota_{\mathcal{V}}(\boldsymbol{X}), \tag{6}$$

where $\mathcal{V} = \{\boldsymbol{X} \in \{0,1\}^{N \times K} : \boldsymbol{X}\mathbf{1} = \mathbf{1}\}$ is the set of row-wise one-hot assignments and $\iota_{\mathcal{V}}$ is its indicator (0 on $\mathcal{V}$, $+\infty$ otherwise). We additionally *enforce nonempty clusters algorithmically* through a simple empty-cluster repair step if they arise, and only require $v_k(\boldsymbol{X}) > 0$ *locally* in the analysis, where

$$v_k(\boldsymbol{X}) = \boldsymbol{x}_k^\top \boldsymbol{D} \boldsymbol{x}_k. \tag{7}$$

On $\{\boldsymbol{X} \in \mathcal{V} : v_k(\boldsymbol{X}) > 0, \forall k\}$, the map $f$ is $C^1$, so $h = -f$ is smooth, while the nonsmooth part $g$ *exactly* encodes one-hot feasibility. No heuristic rounding is needed.

To derive the gradient of the smooth part efficiently, we introduce the cached quantities

$$\boldsymbol{G} = \boldsymbol{W}\boldsymbol{X} \in \mathbb{R}^{N \times K}, \qquad \boldsymbol{q} = (\boldsymbol{X} \odot \boldsymbol{G})^\top \mathbf{1} \in \mathbb{R}^K, \qquad \boldsymbol{v} = \boldsymbol{X}^\top \boldsymbol{d} \in \mathbb{R}^K, \tag{8}$$

so that $q_k = \boldsymbol{x}_k^\top \boldsymbol{W}\boldsymbol{x}_k$ is the intra-cluster association of cluster $k$, and $v_k = \boldsymbol{x}_k^\top \boldsymbol{D}\boldsymbol{x}_k$ is its volume. By the quotient rule,

$$\nabla_{\boldsymbol{x}_k}\left(\frac{q_k}{v_k}\right) = \frac{2}{v_k^2}\left(v_k \boldsymbol{W}\boldsymbol{x}_k - q_k \boldsymbol{D}\boldsymbol{x}_k\right), \quad \Rightarrow \quad \nabla f(\boldsymbol{X}) = 2\left(\boldsymbol{G}\boldsymbol{v}^{-T} - (\boldsymbol{D}\boldsymbol{X})\boldsymbol{\rho}^\top\right), \ \rho_k = \frac{q_k}{v_k^2}. \tag{9}$$

Here $\boldsymbol{v}^{-T}$ denotes column-wise division by $\boldsymbol{v}$.

For the nonsmooth part $g$, Falcon applies a proximal projection onto discrete assignments. Let $\boldsymbol{X}^{(t)}$ be the current iterate. If $\nabla f$ is locally $L_t$-Lipschitz in a neighborhood where $v_k > 0$, then for any $\tau_t \geq L_t$ the standard smoothness inequality gives a local quadratic *minorizer* of $f$:

$$f(\boldsymbol{Y}) \ \geq \ f(\boldsymbol{X}^{(t)}) + \langle \nabla f(\boldsymbol{X}^{(t)}), \boldsymbol{Y} - \boldsymbol{X}^{(t)} \rangle - \frac{\tau_t}{2} \|\boldsymbol{Y} - \boldsymbol{X}^{(t)}\|_F^2. \tag{10}$$

Maximizing this surrogate *over the original discrete set* $\mathcal{V}$ is row-separable. Because each feasible row is one-hot, $\|\boldsymbol{y}_i\|_2^2 \equiv 1$ is constant and $-\frac{\tau_t}{2}\|\boldsymbol{y}_i - \boldsymbol{X}_i^{(t)}\|_2^2 = \frac{\tau_t}{2}(2\langle \boldsymbol{y}_i, \boldsymbol{X}_i^{(t)}\rangle - 1)$, so only the cross term remains. This yields the closed-form row-wise update

$$\boldsymbol{x}_i^{(t+1)} \ = \ e_{\arg\max_{k \in [K]} \ \mu_{ik}^{(t)} + \tau_t X_{ik}^{(t)}}, \qquad \boldsymbol{\mu}^{(t)} = \nabla f(\boldsymbol{X}^{(t)}), \tag{11}$$

where ties are broken deterministically (e.g. by the smallest $k$). Here $\mu_{ik}^{(t)}$ is the first-order gain score of assigning token $i$ to cluster $k$, while the additive term $\tau_t X_{ik}^{(t)}$ acts as an inertia bonus that favors keeping the current label unless another assignment is sufficiently better. Thus, each step remains fully discrete and feasible while avoiding gratuitous label flips.

Because $L_t$ is unknown and may vary, we adopt a monotone line search on $\tau_t$: start with $\tau_t = \tau_0 > 0$; compute equation 11; *accept* if

$$f(\boldsymbol{X}^{(t+1)}) \ \geq \ f(\boldsymbol{X}^{(t)}) \ + \ \frac{\delta\,\tau_t}{2}\|\boldsymbol{X}^{(t+1)} - \boldsymbol{X}^{(t)}\|_F^2, \qquad \delta \in (0,1), \tag{12}$$

otherwise set $\tau_t \leftarrow \gamma\tau_t$ ($\gamma > 1$) and recompute. This Armijo-type sufficient *increase* guarantees monotonic ascent of $f$ and finite termination of the backtracking. Equivalently, in the minimization view for $\Phi(\boldsymbol{X}) = -f(\boldsymbol{X}) + \iota_{\mathcal{V}}(\boldsymbol{X})$, equation 12 is identical to

$$\Phi(\boldsymbol{X}^{(t+1)}) \ \leq \ \Phi(\boldsymbol{X}^{(t)}) \ - \ \frac{\delta\,\tau_t}{2}\|\boldsymbol{X}^{(t+1)} - \boldsymbol{X}^{(t)}\|_F^2,$$

which we use in the Kurdyka–Łojasiewicz based analysis (Appendix C).

**Convergence under the Kurdyka–Łojasiewicz (KL) Property.** Falcon is a forward–backward scheme with monotone backtracking: the update equation 11 is the proximal step for $g = \iota_{\mathcal{V}}$ (row-wise projection), driven by the linearization of $h = -f$, while equation 12 provides a quantitative acceptance rule. Under standard conditions, such as bounded sublevel sets, local Lipschitz continuity of $\nabla f$ in a positive-volume neighborhood, and the KL property of $\Phi$ at a limit point, the generated sequence enjoys: (i) finite termination of backtracking and sufficient descent of $\Phi$; (ii) a uniform subgradient–step bound; (iii) convergence of the whole sequence[2]; and (iv) local linear rates when the KL exponent satisfies $\theta \leq \frac{1}{2}$. The precise assumptions, lemmas, and proof (following Jia et al. (2023), Section 4) are given in Appendix C, with the notational correspondence $\tau_t \leftrightarrow \gamma_k$ and the same $\delta$.

---

[2]Here, "global convergence" refers to the convergence of the entire sequence to a stationary point, rather than convergence to a global optimum of the NP-hard NCut objective.

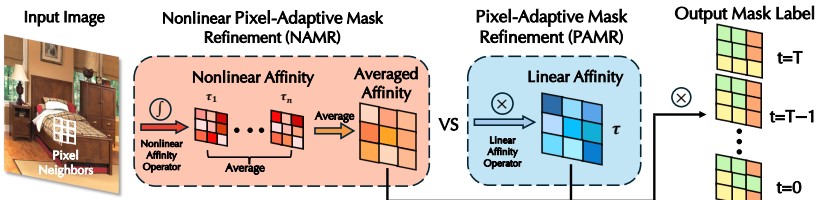

**Figure 2: NAMR vs PAMR.** NAMR computes pixel affinities using a nonlinear operator and averages them across multiple temperatures. In contrast, PAMR uses a linear operator and computes affinities at a single temperature. After affinities are obtained, mask labels are iteratively updated based on them.

Our discrete NCut objective involves polynomial expressions of the affinity matrix together with one-hot constraints on cluster assignments, and is therefore semi-algebraic. Semi-algebraic functions satisfy the KL property Bolte et al. (2007); Attouch et al. (2013). Under the additional condition that the KL exponent satisfies $\theta \leq \frac{1}{2}$, our proximal-gradient scheme enjoys at least a local linear convergence rate near stationary points, in addition to the monotonic descent guaranteed by the backtracking line search.

### 3.3 SEGMENTATION MASK GENERATION

The optimization in Section 3.2 yields a *discrete* token-level assignment $\ell \in \{1, \ldots, K\}^{h \times w}$ on the transformer token grid. While this already provides a coarse segmentation, its spatial resolution $(h, w)$ is much lower than that of the input image. We therefore first perform a lightweight densification on an *intermediate* grid of size $(H_m, W_m)$ (e.g., $128 \times 128$), which improves spatial coherence while remaining computationally inexpensive.

We upsample the discrete label map by nearest-neighbor interpolation to obtain $\hat{\ell} \in \{1, \ldots, K\}^{H_m \times W_m}$. In parallel, the feature map is bilinearly upsampled to $\boldsymbol{Z} \in \mathbb{R}^{C \times H_m \times W_m}$; we denote by $\mathbf{z}_{u,v} \in \mathbb{R}^C$ the feature embedding at location $(u, v)$ on the intermediate grid. The nearest-neighbor step preserves sharp boundaries in the label field, whereas bilinear interpolation maintains smooth variation in the embedding space.

From $\hat{\ell}$ we form the one-hot mask tensor $\boldsymbol{M} \in \{0, 1\}^{K \times H_m \times W_m}$ with $M_{k,u,v} = 1$ iff $\hat{\ell}_{u,v} = k$. The prototype of partition $k$ is the average embedding of its assigned locations:

$$\mathbf{p}_k = \frac{\sum_{u=1}^{H_m} \sum_{v=1}^{W_m} M_{k,u,v}\, \mathbf{z}_{u,v}}{\sum_{u=1}^{H_m} \sum_{v=1}^{W_m} M_{k,u,v}}, \qquad \mathbf{p}_k \in \mathbb{R}^C. \tag{13}$$

These prototypes summarize cluster statistics on the intermediate grid. Each location is then reassigned to the prototype with the largest dot-product similarity:

$$\ell_{u,v}^* = \arg\max_{k \in [K]} \mathbf{z}_{u,v}^\top \mathbf{p}_k, \tag{14}$$

yielding the refined mask $\ell^* \in \{1, \ldots, K\}^{H_m \times W_m}$. This intermediate-resolution densification reduces tokenization artifacts and improves spatial coherence, and can be further upsampled to the original image size if desired.

### 3.4 PIXEL-LEVEL MASK REFINEMENT

Falcon's core contribution is the discrete NCut solver described in Section 3.2. The refinement module is optional and independent of the solver: it is included only to form a complete segmentation pipeline and to align our evaluation protocol with prior work. In particular, Falcon can be paired with no refiner, with PAMR, with DenseCRF, or with other lightweight post-processing methods.

Starting from the intermediate-resolution mask, we optionally recover a pixel-level mask aligned with image boundaries by applying a lightweight refinement step on top of a nearest-neighbor (NN) upsampled initialization. Classical refiners such as DenseCRF can improve boundary adherence, but may introduce additional implementation-dependent overhead or require solving global systems, making them less convenient in modern GPU-heavy pipelines.

**Pixel-Adaptive Mask Refinement (PAMR).** We adopt PAMR as a lightweight edge-aware refiner following prior work. Let $I \in \mathbb{R}^{H \times W \times 3}$ be the RGB image and $M_0 \in [0, 1]^{(C+1) \times H \times W}$ the

NN-upsampled class-probability tensor (including background). For a pixel $(i, j)$ and a local neighborhood $\mathcal{N}(i, j)$ (built from small $3 \times 3$ kernels with dilations), PAMR forms per-direction affinities by a softmax over negative, locally normalized RGB differences:

$$\alpha_{i,j,l,n} = \frac{\exp\left(-\bar{r}(I_{i,j}, I_{l,n})/(\varepsilon + \sigma_{i,j})\right)}{\sum_{(q,r) \in \mathcal{N}(i,j)} \exp\left(-\bar{r}(I_{i,j}, I_{q,r})/(\varepsilon + \sigma_{i,j})\right)}, \quad \bar{r}(I_{i,j}, I_{l,n}) = \tfrac{1}{3}\sum_{k=1}^{3}\left|I_k(i,j) - I_k(l,n)\right|,$$

where $\sigma_{i,j}$ is the local standard deviation of $\bar{r}$ around $(i, j)$ and $\varepsilon > 0$ ensures numerical stability. Refinement proceeds by $T$ iterations of locally weighted averaging:

$$M_{:,i,j}^t = \sum_{(l,n) \in \mathcal{N}(i,j)} \alpha_{i,j,l,n} M_{:,l,n}^{t-1}, \qquad t = 1, \ldots, T, \quad M^0 := M_0.$$

Each update is a convex combination across neighbors (row-stochastic weights), yielding an anisotropic diffusion that respects intensity edges and is stable under small $T$.

**Nonlinear Adaptive Mask Refinement (NAMR).** To show that Falcon is not tied to a specific refinement choice, we also instantiate a simple nonlinear variant, NAMR. NAMR replaces $\bar{r}$ with a contrast-amplified discrepancy $\bar{r}^{\mathrm{nl}}$ obtained by a pointwise nonlinearity $\phi$ (e.g., $\phi(x) = x + 1.5\,\mathrm{ELU}(x)$) before taking absolute values and channel averaging, and aggregates multiple temperature values $\tau \in \mathcal{T}$:

$$\alpha_{i,j,l,n}^{(\tau)} = \mathrm{softmax}_{(l,n) \in \mathcal{N}(i,j)}\left(-\frac{\bar{r}^{\mathrm{nl}}(I_{i,j}, I_{l,n})}{\varepsilon + \tau\,\sigma_{i,j}}\right), \quad M_T^{(\tau)} = \mathsf{MP}_{\alpha^{(\tau)}}^T(M_0), \quad \bar{M}_T = \tfrac{1}{|\mathcal{T}|}\sum_{\tau \in \mathcal{T}} M_T^{(\tau)}.$$

Small $\tau$ emphasizes edge preservation, while larger $\tau$ favors smoothing; averaging across temperatures improves robustness across textures and scales. In experiments, we compare several postprocessing choices—including *NN only*, *PAMR*, and *NAMR* in the main setting, and *DenseCRF* in controlled comparisons—to show that Falcon's gains do not depend on any particular refiner.

## 4 EXPERIMENTS

### 4.1 IMPLEMENTATION DETAILS

We evaluate Falcon in the *training-free* zero-shot setting, where a frozen pretrained encoder is used only at inference time, without mask annotations, task-specific finetuning, or adaptive self-training.

**Datasets and Metrics.** We evaluate Falcon on six widely used benchmarks spanning objects, scenes, and urban environments: Pascal VOC Everingham et al. (2015) (20 classes), Pascal Context Mottaghi et al. (2014) (59 classes with contextual labels), COCO-Object Lin et al. (2014) (80 categories), COCO-Stuff-27 Lin et al. (2014) (27 consolidated classes), Cityscapes Cordts et al. (2016) (27 urban categories), and ADE20K Zhou et al. (2019a) (150 classes). Segmentation quality is measured by mean intersection-over-union (mIoU). Since Falcon produces class-agnostic masks, we align predictions with ground-truth labels using the Hungarian algorithm Kuhn (1955), with a many-to-one mapping for background categories. As a secondary metric, we report Pixel Accuracy (Pixel Acc.), i.e., the proportion of correctly classified pixels after the same alignment procedure.

**Settings.** Following the evaluation protocol of previous work for a fair comparison in the training-free setting, all input images are resized to $1024 \times 1024$. Features are extracted at lower resolutions: $32 \times 32$ from diffusion models Gupta et al. (2024) (prompt-free, $t{=}50$) and $64 \times 64$ from DINOv3 Siméoni et al. (2025), followed by $\ell_2$ normalization and an $\alpha$-power transformation on affinities. Falcon first produces an initial token-level segmentation, which is then densified on an intermediate $128 \times 128$ grid as described in Section 3.4 before any optional pixel-level refinement.

The number of segments $K$ is estimated *before* Falcon optimization via a simple spectral heuristic: we compute the eigenvalues of the normalized affinity matrix once, count eigenmodes below a negative cutoff $\kappa$, and clamp the resulting value to a predefined range. We use a single $\kappa$ per dataset rather than tuning it per image, reflecting dataset-specific annotation granularity. In practice, this spectral step contributes only a small fraction of the total runtime, and Section 4.4 / Figure 4 shows that performance remains stable over a reasonable range of $\kappa$ values.

Unless otherwise noted, Falcon is paired with PAMR in the main setting to align with prior training-free segmentation pipelines; we additionally report no-refinement and NAMR variants, and include

Table 1: **Unsupervised segmentation on six benchmarks (higher is better).** Reports **mIoU**. MaskCLIP Dong et al. (2023) requires text prompts. [†]: reported by us. AutoSC, DiffCut, and Falcon use the SSD-1B encoder in the main setting.

| Method | mIoU (%) ↑ | | | | | |
|---|---|---|---|---|---|---|
| | VOC | Context | COCO-Object | COCO-Stuff-27 | Cityscapes | ADE20K |
| MaskCLIP Dong et al. (2023) | 38.80 | 23.60 | 20.60 | 19.60 | 10.00 | 9.80 |
| MaskCut Wang et al. (2023a) | 53.80 | 43.40 | 30.10 | 41.70 | 18.70 | 35.70 |
| DiffSeg Tian et al. (2024) | 49.80 | 48.80 | 23.20 | 44.20 | 16.80 | 37.70 |
| DiffCut Couairon et al. (2025) | 65.20 | 56.50 | 34.10 | 49.10 | 30.60 | 44.30 |
| AutoSC[†] (ours, refined) | 77.57 | 57.27 | 61.56 | 49.39 | 25.72 | 40.10 |
| DiffCut[†] (ours, refined) | 71.68 | **58.17** | 61.65 | 49.18 | 30.77 | 44.40 |
| **Falcon (ours)** | **78.40** | 57.15 | **61.80** | **50.37** | **33.69** | **45.17** |

controlled DenseCRF comparisons in the appendix. Experiments are run in PyTorch on a single NVIDIA RTX 4090.

**Mask refinement.** Since Falcon's core contribution is the discrete NCut solver, refinement is treated as an optional post-processing component used to align with prior evaluation protocols and to test refiner-agnostic robustness. For **PAMR**, we follow the standard DiffCut setting: $3\times3$ kernels with dilations $D = \{1, 2, 4, 8\}$, yielding $P = 8|D|$ directional filters, $T = 10$ refinement steps, per-pixel local variance $\sigma_{i,j}$ over $\mathcal{N}(i, j)$, and $\varepsilon = 10^{-6}$. Gradients are not propagated through the refiner. For **NAMR**, we use the same neighborhoods and $T = 10$, define $\phi(x) = x + 1.5 \, \text{ELU}(x)$ to construct $\bar{r}^{\text{nl}}$, and average over a temperature set $\mathcal{T} = \{0.06, 0.08, 0.10, 0.12, 0.14, 0.16, 0.18\}$. Unless noted otherwise, no additional data-fidelity mixing with $M_0$ is applied. The **NN** baseline simply upsamples the intermediate mask to $(H, W)$ via nearest-neighbor interpolation without further refinement.

## 4.2 MAIN RESULTS

**Overall performance.** Table 1 reports mIoU on six standard benchmarks. Unless noted otherwise, AutoSC, DiffCut, and Falcon use the SSD-1B encoder with PAMR refinement, matching the setting highlighted in Table 2. Our main comparison focuses on training-free methods, which match Falcon's setting of using frozen pretrained features without additional training.

Relative to the strongest published training-free baseline, DiffCut Couairon et al. (2025), Falcon establishes a new state of the art on five out of six datasets: VOC (+13.2, 78.40 vs. 65.2), COCO-Object (+27.7, 61.80 vs. 34.1), COCO-Stuff-27 (+1.3, 50.37 vs. 49.1), Cityscapes (+3.1, 33.69 vs. 30.6), and ADE20K (+0.9, 45.17 vs. 44.3); on Pascal Context, Falcon remains competitive (57.15 vs. 56.5). These results show that Falcon consistently improves upon the prior training-free state of the art.

We further include refined re-implementations of DiffCut and AutoSC ([†]) under our unified pipeline for a stronger controlled comparison. While these enhanced baselines outperform their official counterparts (e.g., DiffCut[†] reaches 61.65 on COCO-Object and 58.17 on Context), Falcon still surpasses them in most cases: +0.83 on VOC, +0.98 on COCO-Stuff-27, +2.92 on Cityscapes, and +0.77 on ADE20K. Averaged over all six datasets, Falcon attains **54.43** mIoU, exceeding the strongest per-dataset controlled baseline by **+0.77** points on average.

Compared with earlier prompt-free methods (MaskCut, DiffSeg) and stronger controlled re-implementations (DiffCut[†], AutoSC[†]), Falcon delivers robust gains across both object-centric and scene-centric benchmarks, supporting the effectiveness of directly optimizing the discrete $K$-way NCut objective.

**Across encoders and refinement.** Table 2 analyzes robustness across feature backbones (SSD-1B, DINOv3-B, SD2.1) and post-processing choices (none / PAMR / NAMR). SSD-1B yields the strongest overall results and serves as our default setting. Refinement effects are dataset-dependent: on scene-heavy datasets (Cityscapes, COCO-Stuff-27, ADE20K), PAMR and NAMR usually provide additional gains, whereas on object-centric VOC, the raw Falcon segmentation is already very strong (79.15 without refinement). DINOv3-B and SD2.1 trail SSD-1B in absolute performance but preserve the same qualitative trend across post-processing choices.

Table 2: **Unsupervised segmentation across encoders and post-processing choices (higher is better).** Columns report **Pixel Acc. (%)** and **mIoU (%)** for each encoder.

| Benchmark | Method | Refin. | SSD-1B | | DINOv3-B | | SD2.1 | |
|---|---|---|---|---|---|---|---|---|
| | | | Pixel Acc. (%) | mIoU (%) | Pixel Acc. (%) | mIoU (%) | Pixel Acc. (%) | mIoU (%) |
| Cityscapes | DiffCut[†] (ours, refined) | none | 83.55 | 28.35 | 28.38 | 12.75 | 57.29 | 17.09 |
| | | pamr | 85.91 | 30.77 | 29.94 | 12.02 | 67.26 | 19.41 |
| | | namr | 86.02 | 30.95 | 29.48 | 12.37 | 65.40 | 19.41 |
| | Falcon | none | 83.08 | 30.56 | 63.26 | **25.29** | 78.56 | 26.65 |
| | | pamr | 83.74 | **33.69** | 63.85 | 24.40 | 82.81 | 28.04 |
| | | namr | 83.65 | 33.50 | 63.88 | 24.41 | 82.87 | **28.21** |
| VOC | DiffCut[†] (ours, refined) | none | 81.60 | 68.40 | 76.71 | 59.41 | 70.41 | 50.88 |
| | | pamr | 81.81 | 71.68 | 76.66 | 65.73 | 71.57 | 60.41 |
| | | namr | 82.04 | 71.94 | 76.54 | 65.63 | 71.40 | 60.16 |
| | Falcon | none | 88.12 | **79.15** | 84.48 | **76.44** | 86.78 | **77.94** |
| | | pamr | 87.97 | 78.40 | 83.79 | 75.27 | 86.51 | 77.10 |
| | | namr | 88.28 | 78.83 | 84.19 | 75.62 | 86.78 | 77.35 |
| Context | DiffCut[†] (ours, refined) | none | 77.86 | 54.55 | 61.14 | 37.69 | 70.45 | 45.66 |
| | | pamr | 80.36 | **58.17** | 61.16 | 42.63 | 74.02 | 52.05 |
| | | namr | 80.25 | 58.10 | 61.24 | 43.04 | 73.57 | 52.02 |
| | Falcon | none | 78.47 | 54.90 | 62.08 | 44.24 | 75.53 | 52.38 |
| | | pamr | 80.53 | 57.15 | 62.22 | 45.00 | 78.04 | **55.56** |
| | | namr | 80.48 | 57.23 | 62.31 | **45.21** | 77.84 | 55.39 |
| COCO-Stuff | DiffCut[†] (ours, refined) | none | 74.97 | 46.21 | 23.35 | 13.07 | 60.73 | 32.41 |
| | | pamr | 78.47 | 49.18 | 24.43 | 13.54 | 67.37 | 37.43 |
| | | namr | 78.19 | 49.05 | 24.08 | 13.40 | 66.18 | 36.49 |
| | Falcon | none | 75.85 | 47.88 | 48.39 | **30.43** | 72.17 | 43.64 |
| | | pamr | 78.56 | **50.37** | 48.82 | 30.32 | 75.07 | **46.41** |
| | | namr | 78.39 | 50.28 | 48.80 | **30.43** | 74.64 | 46.18 |
| COCO-Object | DiffCut[†] (ours, refined) | none | 80.28 | 62.96 | 72.42 | 52.71 | 67.36 | 48.82 |
| | | pamr | 77.44 | 61.65 | 71.74 | 59.11 | 67.93 | 54.70 |
| | | namr | 78.60 | 62.94 | 71.75 | 60.55 | 67.88 | 55.53 |
| | Falcon | none | 84.67 | **63.98** | 69.55 | **62.19** | 76.17 | 59.21 |
| | | pamr | 81.22 | 61.80 | 67.70 | 59.69 | 73.76 | 58.21 |
| | | namr | 82.30 | 62.74 | 68.31 | 60.78 | 74.68 | **59.41** |
| ADE20K | DiffCut[†] (ours, refined) | none | 69.55 | 42.54 | 28.05 | 26.50 | 60.43 | 38.38 |
| | | pamr | 72.15 | 44.40 | 29.23 | 25.85 | 65.82 | 41.92 |
| | | namr | 71.84 | 44.53 | 28.86 | 26.32 | 64.71 | 41.56 |
| | Falcon | none | 67.64 | 43.05 | 59.15 | 41.87 | 68.33 | 41.11 |
| | | pamr | 70.85 | **45.17** | 60.20 | 42.29 | 70.73 | 42.62 |
| | | namr | 70.38 | 45.06 | 60.04 | **42.44** | 70.42 | **42.70** |

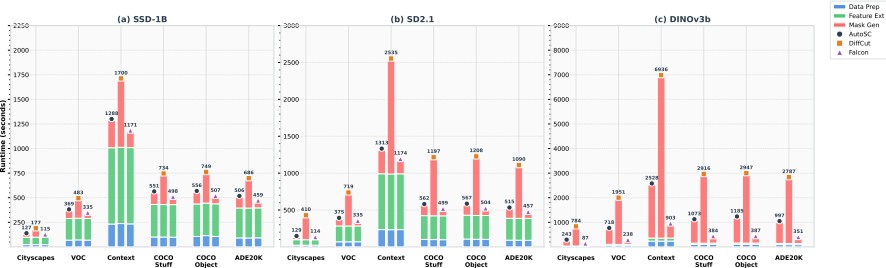

Figure 3: **End-to-end evaluation time across datasets.** Falcon substantially reduces inference time relative to recursive NCut baselines on a single RTX 4090.

Importantly, Falcon consistently outperforms DiffCut across encoders and refiners, showing that its gains come from the solver itself rather than from any particular backbone or post-processing module. This supports the refiner-agnostic design of Falcon and its robustness across diverse pretrained representations.

### 4.3 RUNTIME AND EFFICIENCY

Table 3 and the stacked-bar plots report the end-to-end evaluation time across *Data Preparation*, *Feature Extraction*, and *Mask Generation*. The first two stages are identical across methods, so the runtime differences stem primarily from the graph-partitioning solver. In the main paper, we

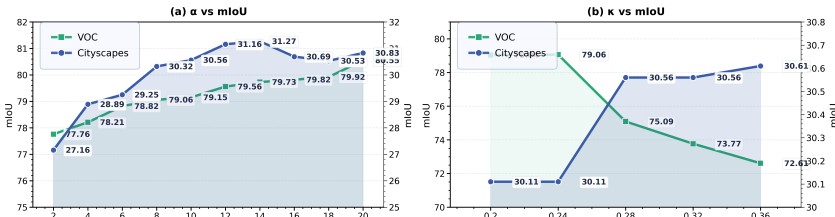

Figure 4: **Hyperparameter sensitivity.** Impact of the power parameter $\alpha$ (left) and spectral cutoff $\kappa$ (right) on mIoU for Cityscapes and VOC.

therefore emphasize end-to-end comparisons, while appendix tables further decompose inference into graph-cut time and refinement time, including controlled comparisons against MaskCut and DenseCRF-based variants.

Falcon consistently delivers the fastest *Mask Generation* time, yielding the lowest overall *Total Time*. The speedup is particularly striking when token counts are high: for example, on *DINOv3-B, Cityscapes*, Falcon reduces Total Time from 784.04s (DiffCut) to 87.47s, and Mask Generation from 747.97s to 52.49s. These results show that Falcon scales much more favorably than recursive NCut solvers as token resolution increases. This strong practical efficiency is also consistent with our convergence analysis: in practice, Falcon typically converges in very few outer iterations, so the monotone proximal scheme translates into a fast solver.

### 4.4 HYPERPARAMETER TUNING STUDIES

**Power Transformation in the Affinity Matrix.** In high-dimensional embedding spaces, graph-based methods often suffer from similarity collapse, where pairwise affinities become nearly uniform and blur true cluster boundaries. To address this, we apply a power transformation that non-linearly rescales affinities: stronger connections are amplified while weaker ones are suppressed. This contrast enhancement sharpens the affinity structure, reduces sensitivity to noise, and yields more stable partitions. As shown in Figure 4, tuning the power parameter $\alpha$ leads to promising gains in mean Intersection-over-Union (mIoU), validating the effectiveness of this transformation for clarifying semantic structure.

**Spectral Threshold for $K$ Selection.** The number of segments $K$ is estimated via a simple spectral heuristic: we compute the eigenvalues of the normalized affinity matrix once, count those below a negative cutoff $\kappa$, and clamp the result to a predefined range. In practice, we use a single $\kappa$ per dataset rather than tuning it per image. This dataset-level choice reflects annotation granularity: object-centric datasets such as VOC typically favor a smaller effective number of segments, whereas denser scene-centric datasets such as Cityscapes or COCO-Stuff tend to favor larger values.

Varying $\kappa$ does affect segmentation quality, but the effect is generally moderate over a semantically reasonable range. As shown in Figure 4, Cityscapes and Pascal VOC exhibit slightly different sensitivities to $\kappa$, yet overall performance remains relatively stable. This suggests that Falcon is robust to the precise cutoff, while still allowing explicit control over segmentation granularity.

## 5 CONCLUSION

We introduced Falcon, a proximal-gradient solver for the discrete $K$-way NCut objective that operates directly on one-hot assignments, avoids spectral relaxation and recursive bipartitioning, preserves feasibility throughout optimization, and enjoys convergence guarantees under the *Kurdyka–Łojasiewicz* framework. Across six benchmarks and three pretrained encoders, Falcon achieves state-of-the-art unsupervised segmentation performance on 17 of 18 benchmark–encoder pairs. It also runs up to an order of magnitude faster than recursive NCut methods and scales more favorably in memory at high resolution, making it practical for larger token graphs in training-free segmentation. These results highlight the value of treating NCut as a principled discrete optimization problem rather than a sequence of relax-and-round heuristics. More broadly, Falcon suggests that combining foundation-model features with efficient discrete solvers is a promising direction for training-free segmentation, pseudo-mask generation, and other structured grouping problems beyond semantic segmentation.

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

# A    EQUIVALENCE OF $K$-WAY NCUT AND A SUM OF GENERALIZED RAYLEIGH QUOTIENTS

We consider an undirected weighted graph with a symmetric nonnegative affinity matrix $\boldsymbol{W} \in \mathbb{R}^{N \times N}$ and degree matrix $\boldsymbol{D} = \mathrm{diag}(\boldsymbol{W}\boldsymbol{1})$, and let $\boldsymbol{L} = \boldsymbol{D} - \boldsymbol{W}$ be the combinatorial Laplacian. A $K$-way partition $\{P_1, \ldots, P_K\}$ is represented by one-hot indicator columns $\boldsymbol{x}_k \in \{0,1\}^N$ and the assignment matrix $\boldsymbol{X} = [\boldsymbol{x}_1, \ldots, \boldsymbol{x}_K] \in \{0,1\}^{N \times K}$ satisfying $\boldsymbol{X}\boldsymbol{1} = \boldsymbol{1}$.

For each part $P_k$, denote its volume and cut by

$$\mathrm{vol}(P_k) = \sum_{i \in P_k} d_i, \qquad \mathrm{cut}(P_k, \bar{P}_k) = \sum_{i \in P_k, \, j \notin P_k} W_{ij}. \tag{15}$$

The normalized cut objective Shi & Malik (2000b) is

$$\mathrm{Ncut}(P_1, \ldots, P_K) = \sum_{k=1}^{K} \frac{\mathrm{cut}(P_k, \bar{P}_k)}{\mathrm{vol}(P_k)}. \tag{16}$$

With the indicator $\boldsymbol{x}_k$ of $P_k$, we have the following standard equalities:

$$\mathrm{vol}(P_k) = \boldsymbol{x}_k^\top \boldsymbol{D} \boldsymbol{x}_k, \qquad \mathrm{cut}(P_k, \bar{P}_k) = \boldsymbol{x}_k^\top \boldsymbol{L} \boldsymbol{x}_k = \boldsymbol{x}_k^\top \boldsymbol{D} \boldsymbol{x}_k - \boldsymbol{x}_k^\top \boldsymbol{W} \boldsymbol{x}_k. \tag{17}$$

Since $[\boldsymbol{x}_k]_i = 1$ iff $i \in P_k$, $\boldsymbol{x}_k^\top \boldsymbol{D} \boldsymbol{x}_k = \sum_{i=1}^{N} d_i [\boldsymbol{x}_k]_i^2 = \sum_{i \in P_k} d_i = \mathrm{vol}(P_k)$. Moreover, using the well-known Laplacian quadratic form $\boldsymbol{z}^\top \boldsymbol{L} \boldsymbol{z} = \frac{1}{2} \sum_{i,j} W_{ij}(z_i - z_j)^2$ and taking $\boldsymbol{z} = \boldsymbol{x}_k \in \{0,1\}^N$, the summand is 1 iff $\{i, j\}$ crosses the cut $(P_k, \bar{P}_k)$, which yields $\boldsymbol{x}_k^\top \boldsymbol{L} \boldsymbol{x}_k = \mathrm{cut}(P_k, \bar{P}_k)$. The decomposition $\boldsymbol{L} = \boldsymbol{D} - \boldsymbol{W}$ gives the alternative expression.

Dividing equation 17 termwise gives

$$\frac{\mathrm{cut}(P_k, \bar{P}_k)}{\mathrm{vol}(P_k)} = \frac{\boldsymbol{x}_k^\top \boldsymbol{L} \boldsymbol{x}_k}{\boldsymbol{x}_k^\top \boldsymbol{D} \boldsymbol{x}_k} = 1 - \frac{\boldsymbol{x}_k^\top \boldsymbol{W} \boldsymbol{x}_k}{\boldsymbol{x}_k^\top \boldsymbol{D} \boldsymbol{x}_k}. \tag{18}$$

Summing equation 18 over $k = 1, \ldots, K$ and using equation 16 yields

$$\mathrm{Ncut}(\boldsymbol{X}) = \sum_{k=1}^{K} \frac{\boldsymbol{x}_k^\top \boldsymbol{L} \boldsymbol{x}_k}{\boldsymbol{x}_k^\top \boldsymbol{D} \boldsymbol{x}_k} = K - \sum_{k=1}^{K} \frac{\boldsymbol{x}_k^\top \boldsymbol{W} \boldsymbol{x}_k}{\boldsymbol{x}_k^\top \boldsymbol{D} \boldsymbol{x}_k}. \tag{19}$$

Define the normalized association

$$f(\boldsymbol{X}) = \sum_{k=1}^{K} \frac{\boldsymbol{x}_k^\top \boldsymbol{W} \boldsymbol{x}_k}{\boldsymbol{x}_k^\top \boldsymbol{D} \boldsymbol{x}_k}, \qquad \text{so that} \qquad \mathrm{Ncut}(\boldsymbol{X}) = K - f(\boldsymbol{X}). \tag{20}$$

Each summand of $f(\boldsymbol{X})$ is a generalized Rayleigh quotient of the matrix pencil $(\boldsymbol{W}, \boldsymbol{D})$ evaluated at the indicator $\boldsymbol{x}_k$. With $K$ fixed, minimizing $\mathrm{Ncut}$ is equivalent to maximizing $f$ over the discrete assignment set $\mathcal{X} := \{\boldsymbol{X} \in \{0,1\}^{N \times K} : \boldsymbol{X}\boldsymbol{1} = \boldsymbol{1}\}$. The set $\mathcal{X}$ is combinatorial and nonconvex, and the ratio structure in equation 19 and equation 20 further complicates the landscape, so the exact $K$-way problem is NP-hard in general. This motivates spectral relaxations and continuous surrogates followed by a discrete projection back onto $\mathcal{X}$.

## B ON THE SUBOPTIMALITY OF RECURSIVE NORMALIZED CUT

Recursive partitioning applies a two-way Normalized Cut (Ncut) repeatedly until $K$ parts are obtained. The procedure is simple and often efficient, yet it may be far from the global optimum of the $K$-way objective. This note explains why greedy bipartitions can deviate from the best $K$-way solution and highlights the mathematical and structural causes.

Let $\mathcal{G} = (\mathcal{V}, \mathcal{E}, \boldsymbol{W})$ be an undirected graph with $|\mathcal{V}| = N$, affinity matrix $\boldsymbol{W} \in \mathbb{R}_+^{N \times N}$, and degree matrix $\boldsymbol{D} = \mathrm{diag}(\boldsymbol{W}\boldsymbol{1})$. A $K$-way partition $\{\mathcal{A}_k\}_{k=1}^K$ minimizes

$$\mathrm{Ncut}(\{\mathcal{A}_k\}) = \sum_{k=1}^K \frac{\mathrm{cut}(\mathcal{A}_k, \mathcal{V} \setminus \mathcal{A}_k)}{\mathrm{vol}(\mathcal{A}_k)}, \qquad \mathrm{cut}(A, B) = \sum_{i \in A,\ j \in B} W_{ij}, \quad \mathrm{vol}(A) = \sum_{i \in A} d_i.$$

Exact minimization is NP-hard for $K > 2$. A common heuristic first finds a two-way Ncut $(\mathcal{A}, \mathcal{B})$ on $\mathcal{G}$, then recurses on the induced subgraphs until $K$ clusters are formed. Despite its convenience, this scheme does not, in general, recover a globally optimal $K$-way partition.

**Lack of optimal substructure.** The $K$-way objective does not decompose into independent two-way subproblems. Once a boundary between $\mathcal{A}$ and $\mathcal{B}$ is fixed, subsequent decisions are constrained within induced subgraphs. A locally optimal bipartition $\mathrm{Ncut}(\mathcal{A}, \mathcal{B})$ need not be part of any globally optimal $K$-way solution.

**Objective mismatch under recursion.** Consider $K = 3$ with ground-truth communities $A, B, C$. If recursion first splits $S := A \cup B$ from $C$, then refines $S$ into $A$ and $B$, the recursive score equals

$$\underbrace{\frac{\mathrm{cut}(S, C)}{\mathrm{vol}(S)} + \frac{\mathrm{cut}(S, C)}{\mathrm{vol}(C)}}_{\text{first bipartition on } \mathcal{G}} + \underbrace{\frac{\mathrm{cut}_S(A, B)}{\mathrm{vol}_S(A)} + \frac{\mathrm{cut}_S(A, B)}{\mathrm{vol}_S(B)}}_{\text{second bipartition on the induced subgraph } S},$$

where $\mathrm{cut}_S$ and $\mathrm{vol}_S$ are computed in the induced subgraph on $S$. By contrast, the global 3-way objective is

$$\frac{\mathrm{cut}(A, B \cup C)}{\mathrm{vol}(A)} + \frac{\mathrm{cut}(B, A \cup C)}{\mathrm{vol}(B)} + \frac{\mathrm{cut}(C, A \cup B)}{\mathrm{vol}(C)}.$$

These expressions differ in the normalizers for the second stage: typically $\mathrm{vol}_S(A) < \mathrm{vol}(A)$ and $\mathrm{vol}_S(B) < \mathrm{vol}(B)$ whenever $A$ or $B$ has edges to $C$. Hence the recursive refinement overweights $\mathrm{cut}_S(A, B) = \mathrm{cut}(A, B)$ by dividing through smaller denominators, which can increase the total objective. There exist graphs where this "normalization gap" makes the recursive sum strictly larger than the global 3-way optimum, and the same phenomenon extends to $K > 3$.

**Irreversibility and greedy commitment.** Early boundaries are irrevocable. A bipartition that reduces the two-way score may merge distinct communities because the two-way objective favors grouping parts whose union has a large volume, even if a later $K$-way split would separate them. Once merged, separating them inside the induced subgraph ignores edges to the rest of the graph, which distorts normalization and may prevent recovery of the best global arrangement.

**Spectral information loss in two-way splits.** Two-way Ncut is driven by the Fiedler vector of $\boldsymbol{L}_{\mathrm{sym}}$. For $K > 2$, higher eigenvectors encode additional community structure. Sequentially applying a single eigenvector per split neglects joint information in the top $K$ eigenvectors and can miss multi-community signals; rounding and recursion cannot, in general, reconstruct the simultaneous $K$-way structure.

Recursive bipartitioning is a useful heuristic, yet it imposes a sequential search on a global objective. Local optimality at each step is not sufficient for global optimality, especially on non-hierarchical graphs or when multiple spectral components are essential. Methods that optimize a $K$-way objective directly or allow global refinement can mitigate these failures.

## C KURDYKA–ŁOJASIEWICZ CONVERGENCE ANALYSIS OF FALCON

Following the Kurdyka–Łojasiewicz (KL) Convergence analysis method in Jia et al. (2023), we establish the convergence of the entire sequence generated by Falcon to a stationary point of the

objective function. This is achieved under the condition that an accumulation point of the sequence satisfies the Kurdyka–Łojasiewicz (KL) property.

## C.1 PROBLEM, ALGORITHM, AND STANDING ASSUMPTIONS

Let $\mathbb{X} = \mathbb{R}^{N \times K}$ with Frobenius inner product and norm $\|\cdot\|$. We consider

$$\min_{\mathbf{X} \in \mathbb{X}} \ \psi(\mathbf{X}) := f(\mathbf{X}) + \phi(\mathbf{X}), \tag{21}$$

where $f : \mathbb{X} \to \mathbb{R}$ is $C^1$ and $\phi : \mathbb{X} \to (-\infty, +\infty]$ is proper, lower semicontinuous (lsc). In FALCON, $\phi = \iota_{\mathcal{V}_+}$ encodes the row-wise one-hot feasibility with nondegenerate cluster volumes; thus $\operatorname{dom}(\phi) = \mathcal{V}_+$.

In Falcon with monotone backtracking, given $\mathbf{X}^0 \in \operatorname{dom}(\phi)$ and parameters

$$\underline{\gamma} > 0, \qquad \eta > 1, \qquad \delta \in (0, 1),$$

for $k = 0, 1, 2, \dots$ perform:

*(A1) Backtracking loop.* Initialize a trial stepsize $\gamma_k \in [\underline{\gamma}, +\infty)$.

*(A2) Forward–backward subproblem.* Compute

$$\mathbf{X}^{k+1} \in \arg\min_{\mathbf{Y} \in \mathbb{X}} \left\langle \nabla f(\mathbf{X}^k), \mathbf{Y} - \mathbf{X}^k \right\rangle + \frac{\gamma_k}{2} \|\mathbf{Y} - \mathbf{X}^k\|^2 + \phi(\mathbf{Y}). \tag{22}$$

When $\phi = \iota_{\mathcal{V}_+}$, equation 22 reduces to row-wise one-hot projection of $\mathbf{X}^k - \gamma_k^{-1} \nabla f(\mathbf{X}^k)$ onto $\mathcal{V}_+$.

*(A3) Acceptance test.* Accept if

$$\psi(\mathbf{X}^{k+1}) \leq \psi(\mathbf{X}^k) - \frac{\delta \gamma_k}{2} \|\mathbf{X}^{k+1} - \mathbf{X}^k\|^2. \tag{23}$$

Otherwise set $\gamma_k \leftarrow \eta \gamma_k$ and repeat (A2)–(A3).

**Assumption C.1** (Standing; cf. Assumption 3.2)**.**

- (a) $\psi$ is bounded from below on $\operatorname{dom}(\phi)$.

- (b) $\phi$ is bounded from below by an affine function.

- (c) $\nabla f$ is locally Lipschitz continuous.

We also assume the subproblem equation 22 is solvable for all $k$ (true here since $\phi$ is an indicator and the objective in $\mathbf{Y}$ is strongly coercive).

**Definition C.2** (KL property (at $\mathbf{X}^\star$))**.** A proper lsc $\psi$ has the Kurdyka–Łojasiewicz (KL) property at $\mathbf{X}^\star$ if there exist $\eta > 0$, a neighborhood $\mathcal{U}$ of $\mathbf{X}^\star$, and a concave $C^1$ function $\chi : (0, \eta) \to \mathbb{R}_+$ with $\chi'(t) > 0$ such that

$$\chi'\big(\psi(\mathbf{X}) - \psi(\mathbf{X}^\star)\big) \cdot \operatorname{dist}\big(0, \partial\psi(\mathbf{X})\big) \ \geq \ 1$$

for all $\mathbf{X} \in \mathcal{U}$ with $\psi(\mathbf{X}^\star) < \psi(\mathbf{X}) < \psi(\mathbf{X}^\star) + \eta$. If $\chi(t) = c\, t^{1-\theta}$ with $\theta \in [0, 1)$, then $\theta$ is the KL exponent.

## C.2 PRELIMINARIES: DESCENT AND OPTIMALITY RESIDUAL

**Lemma C.3** (Finite termination of backtracking and sufficient decrease)**.** *Under Assumption C.1, the backtracking loop terminates after finitely many inner steps. The accepted $(\mathbf{X}^{k+1}, \gamma_k)$ satisfies equation 23. Consequently, $\{\psi(\mathbf{X}^k)\}$ is monotonically decreasing and bounded below; hence it converges.*

*Proof.* By local Lipschitz continuity, there exists $L_k < \infty$ with

$$f(\mathbf{Y}) \leq f(\mathbf{X}^k) + \langle \nabla f(\mathbf{X}^k), \mathbf{Y} - \mathbf{X}^k \rangle + \frac{L_k}{2} \|\mathbf{Y} - \mathbf{X}^k\|^2$$

for $\mathbf{Y}$ near $\mathbf{X}^k$. Combining this with the optimality of equation 22 yields equation 23 whenever $\gamma_k \geq L_k/(1 - \delta)$. Since $\gamma_k$ increases geometrically in backtracking, acceptance occurs in finitely many inner steps. The monotonicity and the lower bound of $\psi$ (Assumption C.1(a)) imply convergence of $\psi(\mathbf{X}^k)$. $\qquad\square$

**Lemma C.4** (Optimality of the prox subproblem and residual bound). *For the accepted* $(\mathbf{X}^{k+1}, \gamma_k)$ *one has*

$$0 \in \nabla f(\mathbf{X}^k) + \gamma_k(\mathbf{X}^{k+1} - \mathbf{X}^k) + \partial\phi(\mathbf{X}^{k+1}). \tag{24}$$

*Hence*

$$\mathrm{dist}\big(0, \partial\psi(\mathbf{X}^{k+1})\big) \;\leq\; \|\nabla f(\mathbf{X}^{k+1}) - \nabla f(\mathbf{X}^k)\| + \gamma_k \|\mathbf{X}^{k+1} - \mathbf{X}^k\|. \tag{25}$$

*Proof.* The inclusion equation 24 is the first-order condition of equation 22. Adding and subtracting $\nabla f(\mathbf{X}^{k+1})$ and using $\partial\psi(\mathbf{X}^{k+1}) = \nabla f(\mathbf{X}^{k+1}) + \partial\phi(\mathbf{X}^{k+1})$ gives equation 25. □

STEP 1 (LEMMA 4.1 ANALOGUE): LOCAL BOUNDEDNESS OF ACCEPTED STEPSIZES

**Lemma C.5** (Local boundedness of $\gamma_k$). *Let* $\mathbf{X}^\star$ *be an accumulation point of* $\{\mathbf{X}^k\}$. *Then there exist a neighborhood* $\mathcal{U}$ *of* $\mathbf{X}^\star$ *and a constant* $\overline{\gamma} < \infty$ *such that, for all sufficiently large* $k$ *with* $\mathbf{X}^k \in \mathcal{U}$, *the accepted stepsize satisfies* $\gamma_k \leq \overline{\gamma}$.

*Proof.* By Assumption C.1(c), $\nabla f$ is $L_\rho$-Lipschitz on a ball $\mathcal{U} = \{\mathbf{Z} : \|\mathbf{Z} - \mathbf{X}^\star\| \leq \rho\}$ for small $\rho > 0$. The descent argument in Lemma C.3 shows that equation 23 holds for any $\gamma \geq L_\rho/(1-\delta)$, provided $\mathbf{X}^k, \mathbf{X}^{k+1} \in \mathcal{U}'$ with slightly larger radius. Thus, once $k$ is large so that iterates stay in $\mathcal{U}'$, the backtracking will accept a step with $\gamma_k \leq \overline{\gamma} := \eta\, L_\rho/(1-\delta)$. □

STEP 2 (LEMMA 4.2 ANALOGUE): CONVERGENCE OF FUNCTION VALUES

**Lemma C.6** ($\psi(\mathbf{X}^k) \to \psi(\mathbf{X}^\star)$). *Let* $\mathbf{X}^\star$ *be an accumulation point of* $\{\mathbf{X}^k\}$. *Then* $\psi(\mathbf{X}^k) \to \psi(\mathbf{X}^\star)$.

*Proof.* By Lemma C.3, $\psi(\mathbf{X}^k)$ converges to some $\psi^\infty$. Take $k_j \to \infty$ with $\mathbf{X}^{k_j} \to \mathbf{X}^\star$. Lower semicontinuity of $\phi$ and continuity of $f$ give $\psi(\mathbf{X}^\star) \leq \liminf_j \psi(\mathbf{X}^{k_j}) = \psi^\infty$. Since $\psi(\mathbf{X}^k)$ is decreasing and $\mathbf{X}^\star$ lies in the same sublevel set, $\psi^\infty \leq \psi(\mathbf{X}^\star)$. Hence equality holds. □

STEP 3 (LEMMA 4.4 ANALOGUE): SUBGRADIENT–STEP RELATION

**Lemma C.7** (Subgradient controlled by step length). *There exist* $k_0$ *and* $C > 0$ *such that for all* $k \geq k_0$,

$$\mathrm{dist}\big(0, \partial\psi(\mathbf{X}^{k+1})\big) \;\leq\; C\, \|\mathbf{X}^{k+1} - \mathbf{X}^k\|. \tag{26}$$

*Proof.* Restrict to the tail where $\mathbf{X}^k$ lies in the neighborhood $\mathcal{U}$ of Lemma C.5. Then $\nabla f$ is $L_\rho$-Lipschitz and $\gamma_k \leq \overline{\gamma}$. From equation 25,

$$\mathrm{dist}\big(0, \partial\psi(\mathbf{X}^{k+1})\big) \leq (L_\rho + \overline{\gamma})\, \|\mathbf{X}^{k+1} - \mathbf{X}^k\|.$$

Set $C := L_\rho + \overline{\gamma}$ and choose $k_0$ large so that the neighborhood assumptions hold. □

## C.3 MAIN THEOREM (THEOREM 4.5 ANALOGUE): GLOBAL CONVERGENCE

**Theorem C.8** (Global convergence and finite length). *Suppose Assumption C.1 holds and the sequence* $\{\mathbf{X}^k\}$ *generated by* FALCON *has an accumulation point* $\mathbf{X}^\star$ *at which* $\psi$ *satisfies the KL property. Then* $\{\mathbf{X}^k\}$ *converges to* $\mathbf{X}^\star$ *and*

$$\sum_{k=0}^{\infty} \|\mathbf{X}^{k+1} - \mathbf{X}^k\| \;<\; \infty.$$

*Proof.* Let $\Delta_k := \psi(\mathbf{X}^k) - \psi(\mathbf{X}^\star) > 0$. By Lemma C.3 and Lemma C.6, $\Delta_k \downarrow 0$. By Lemma C.7, for $k \geq k_0$,

$$\mathrm{dist}\big(0, \partial\psi(\mathbf{X}^k)\big) \;\leq\; C\, \|\mathbf{X}^k - \mathbf{X}^{k-1}\|.$$

The KL inequality at $\mathbf{X}^\star$ yields

$$\chi'(\Delta_k) \;\geq\; \frac{1}{C\,\|\mathbf{X}^k - \mathbf{X}^{k-1}\|} \qquad (k \geq k_0). \tag{27}$$

From the sufficient decrease equation 23 and $\gamma_k \geq \underline{\gamma}$,

$$\Delta_k - \Delta_{k+1} \; \geq \; \frac{\delta\,\underline{\gamma}}{2}\,\|\mathbf{X}^{k+1} - \mathbf{X}^k\|^2.$$

Concavity of $\chi$ implies

$$\chi(\Delta_k) - \chi(\Delta_{k+1}) \; \geq \; \chi'(\Delta_k)\,(\Delta_k - \Delta_{k+1}) \; \geq \; \frac{\delta\,\underline{\gamma}}{2C}\,\frac{\|\mathbf{X}^{k+1} - \mathbf{X}^k\|^2}{\|\mathbf{X}^k - \mathbf{X}^{k-1}\|}.$$

Summing from $k = k_0$ to $m$ and telescoping the left-hand side gives

$$\sum_{k=k_0}^{m} \frac{\|\mathbf{X}^{k+1} - \mathbf{X}^k\|^2}{\|\mathbf{X}^k - \mathbf{X}^{k-1}\|} \; \leq \; \frac{2C}{\delta\,\underline{\gamma}}\big(\chi(\Delta_{k_0}) - \chi(\Delta_{m+1})\big) \; \leq \; \frac{2C}{\delta\,\underline{\gamma}}\,\chi(\Delta_{k_0}).$$

An elementary inequality then yields $\sum_{k\geq k_0} \|\mathbf{X}^{k+1} - \mathbf{X}^k\| < \infty$ (finite length). Hence $\{\mathbf{X}^k\}$ is Cauchy and converges; as $\mathbf{X}^\star$ is an accumulation point, the whole sequence converges to $\mathbf{X}^\star$. $\quad\square$

## C.4 RATE (THEOREM 4.6 ANALOGUE): LINEAR CONVERGENCE FOR KL EXPONENT $\theta = \frac{1}{2}$

**Theorem C.9** (Linear rates for $\theta = \frac{1}{2}$). *Under the assumptions of Theorem C.8, suppose $\chi(t) = c\,t^{1/2}$ near $\mathbf{X}^\star$ (i.e., KL exponent $\theta = \frac{1}{2}$). Then there exist $q \in (0,1)$, $\omega > 0$, $\mu \in (0,1)$, and $k_1$ such that*

$$\begin{aligned}
\text{(Q-linear values)} \quad & \psi(\mathbf{X}^{k+1}) - \psi(\mathbf{X}^\star) \; \leq \; q\big(\psi(\mathbf{X}^k) - \psi(\mathbf{X}^\star)\big), \quad \forall k \geq k_1, \\
\text{(R-linear iterates)} \quad & \|\mathbf{X}^k - \mathbf{X}^\star\| \; \leq \; \omega\,\mu^k, \quad \forall k \geq k_1.
\end{aligned}$$

*Proof.* With $\chi(t) = c\,t^{1/2}$, equation 27 becomes

$$\frac{c}{2}\,(\Delta_k)^{-1/2} \; \geq \; \frac{1}{C\,\|\mathbf{X}^k - \mathbf{X}^{k-1}\|}.$$

Combining with $\Delta_k - \Delta_{k+1} \geq \frac{\delta\,\underline{\gamma}}{2}\|\mathbf{X}^{k+1} - \mathbf{X}^k\|^2$ and following the standard argument in the proof of Theorem 4.6 (cf. Section 4), one obtains a linear contraction $\Delta_{k+1} \leq q\,\Delta_k$ for all large $k$ and some $q \in (0,1)$. The R-linear rate for iterates follows from summing the sufficient decrease and bounding $\|\mathbf{X}^{k+1} - \mathbf{X}^k\|^2$ in terms of $\Delta_k - \Delta_{k+1}$ to dominate the tail distance to $\mathbf{X}^\star$ by a geometric series. $\quad\square$

## C.5 REMARKS SPECIFIC TO FALCON

**Feasible set and KL property.** When $\phi = \iota_{\mathcal{V}_+}$ and $f$ is analytic in a neighborhood of a limit point $\mathbf{X}^\star \in \mathcal{V}_+$ with nondegenerate cluster volumes, $\psi = f + \iota_{\mathcal{V}_+}$ is definable in an o-minimal structure, thus satisfies the KL property at $\mathbf{X}^\star$.

**Local Lipschitz of $\nabla f$ in token-NCut.** For the normalized association objective used by FALCON, the explicit gradient formula shows $\nabla f$ is locally Lipschitz on any region where per-cluster volumes are bounded away from zero, which is enforced by $\operatorname{dom}(\phi) = \mathcal{V}_+$. This verifies Assumption C.1(c) near accumulation points.

*Template note.* The structure of Lemmas C.5–C.7 and Theorems C.8–C.9 follows the Section 4 framework (descent, subgradient bound, KL) of Jia et al. (2023).

## D TIME COMPLEXITY OF FALCON

We analyze one proximal update of FALCON under an affinity matrix $\mathbf{W} \in \mathbb{R}^{N \times N}$. Let $\mathbf{X} \in \{0,1\}^{N \times K}$ be row-one-hot. In one accepted outer iteration, the algorithm forms

$$\mathbf{G} = \mathbf{W}\mathbf{X}, \qquad \mathbf{q} = (\mathbf{X} \odot \mathbf{G})^\top \mathbf{1}, \qquad \mathbf{v} = \mathbf{X}^\top \mathbf{d}, \qquad \mathbf{d} = \mathbf{W}\mathbf{1}, \qquad \mathbf{D} = \operatorname{diag}(\mathbf{d}),$$

and computes the gradient

$$\nabla f(\boldsymbol{X}) = 2\big(\boldsymbol{G}\,\boldsymbol{v}^{-T} - (\boldsymbol{D}\boldsymbol{X})\,\boldsymbol{\rho}^{\top}\big), \qquad \rho_k = \frac{q_k}{v_k^2}.$$

It then performs the row-wise $\arg\max$ update and evaluates the Armijo-type acceptance criterion. The dominant costs are

$$\text{Per iteration cost} \;=\; \underbrace{\boldsymbol{G} = \boldsymbol{W}\boldsymbol{X}}_{\mathcal{O}(N^2 K)}$$

$$+ \underbrace{\text{column scalings and subtraction in } \nabla f}_{\mathcal{O}(NK)}$$

$$+ \underbrace{\text{row-wise } \arg\max}_{\mathcal{O}(NK)}$$

$$+ \underbrace{\boldsymbol{G}^{+} = \boldsymbol{W}\boldsymbol{X}^{+}}_{\mathcal{O}(N^2 K)}.$$

Hence one accepted iteration costs $\mathcal{O}(N^2 K)$, omitting lower-order $\mathcal{O}(NK)$ terms.

If $B_t \geq 1$ denotes the number of line-search trials in iteration $t$ (including the accepted one), each trial repeats $\boldsymbol{W}\boldsymbol{X}^{+}$ together with light $\mathcal{O}(NK)$ work, so the cost of iteration $t$ is $\mathcal{O}(B_t N^2 K)$. Near the limit set, Armijo acceptance holds for any $\tau \geq L_\rho/(1-\delta)$; starting from $\tau_0 > 0$ and multiplying by $\gamma > 1$ gives

$$B_t \;\leq\; 1 + \Big\lceil \log_\gamma\Big(\frac{L_\rho}{(1-\delta)\tau_0}\Big)\Big\rceil,$$

i.e., a uniform *constant* bound, so backtracking contributes only a small multiplicative factor. If $B_t = \mathcal{O}(1)$ and $T$ accepted outer iterations are performed, then

$$\text{Total cost} \;=\; \mathcal{O}\Big(\big(\textstyle\sum_{t=1}^{T} B_t\big) N^2 K\Big) \;=\; \mathcal{O}(T N^2 K), \qquad T, K \ll N.$$

Classical spectral methods (eigenvector computation for the Laplacian or normalized affinity) and recursive NCut typically require solving an $N \times N$ eigenproblem, leading to cubic $\mathcal{O}(N^3)$ time in standard dense linear algebra. In contrast, Falcon's core cost is quadratic $\mathcal{O}(N^2)$ for constant $K$ and $T$. Moreover, our KL-based analysis guarantees convergence of the whole sequence (and linear rates when the KL exponent is $1/2$), a property that plain power or eigen-solvers used in recursive NCut do not directly provide at the discrete assignment level.

## E  PROPERTIES AND TIME COMPLEXITY OF PAMR AND NAMR

Let $\Omega \subset \mathbb{Z}^2$ be the pixel grid, $|\Omega| = N$, with image $I : \Omega \to \mathbb{R}^3$. The initial segmentation is $M_0 \in [0,1]^{(C+1)\times N}$ (one-vs-rest probabilities from NN-upsampled token masks). For a set of dilations $D \subset \mathbb{N}^{+}$, define displacement sets $\Delta_d$ of the 8 neighbors at distance $d$ and $\Delta = \bigcup_{d\in D} \Delta_d$ with $P = 8|D|$. For $\delta \in \Delta$, the shift operator $(S_\delta x)(i) = x(i+\delta)$ uses border replication.

For each $(i,\delta)$, define channelwise discrepancies

$$r_k(i,\delta) = \big|I_k(i) - I_k(i+\delta)\big|, \quad \bar{r}(i,\delta) = \tfrac{1}{3}\sum_{k=1}^{3} r_k(i,\delta), \quad \sigma(i) = \mathrm{Std}_{\delta\in\Delta}\,\bar{r}(i,\delta).$$

NAMR replaces $\bar{r}$ by $\bar{r}^{\mathrm{nl}}(i,\delta) = \tfrac{1}{3}\sum_k \big|\phi(I_k(i) - I_k(i+\delta))\big|$ with a fixed pointwise $\phi$.

**PAMR as anisotropic diffusion on a row-stochastic graph.** For a stabilization constant $\varepsilon > 0$ and (optionally) a scale factor $\lambda > 0$ absorbed into $\sigma$, define directional logits and softmax weights

$$\ell(i,\delta) = -\frac{\bar{r}(i,\delta)}{\varepsilon + \sigma(i)}, \qquad p(i,\delta) = \frac{\exp\{\ell(i,\delta)\}}{\sum_{\delta'\in\Delta}\exp\{\ell(i,\delta')\}}.$$

Collect $p(i, \delta)$ into a sparse row-stochastic matrix $W \in \mathbb{R}^{N \times N}$ with $W(i, i+\delta) = p(i, \delta)$ and zeros elsewhere. For each class $c$,

$$m_{t+1}^{(c)} = W \, m_t^{(c)}, \qquad t = 0, \ldots, T-1, \qquad M^0 := M_0, \quad M^T = [m_T^{(1)} \; \cdots \; m_T^{(C+1)}].$$

*Stability.* Since $W$ is row-stochastic, each update is a convex combination, $\|m_{t+1}^{(c)}\|_\infty \leq \|m_t^{(c)}\|_\infty$.
*Energy view (fixed $W$).* A finite number of Jacobi steps approximates minimizing

$$E(M) = \tfrac{1}{2} \sum_{i,j} W(i,j) \, \|M(:,i) - M(:,j)\|_2^2 \; + \; \alpha \, \|M - M_0\|_2^2,$$

with an optional data-fidelity $\alpha \geq 0$. Early stopping ($T \approx 5$–$10$) balances denoising and edge preservation.

**NAMR: non-linear contrast and multi-temperature aggregation.** For $\mathcal{T} = \{\tau_0, \ldots, \tau_{m-1}\}$,

$$\ell^{(\tau)}(i, \delta) = -\frac{\bar{r}^{\mathrm{nl}}(i, \delta)}{\varepsilon + \tau \, \sigma(i)}, \quad p^{(\tau)}(i, \delta) = \mathrm{softmax}_{\delta \in \Delta} \, \ell^{(\tau)}(i, \delta), \quad W^{(\tau)}(i, i+\delta) = p^{(\tau)}(i, \delta).$$

Run $T$ steps per temperature,

$$M_T^{(\tau)} = (W^{(\tau)})^T M_0, \qquad \bar{M}_T = \tfrac{1}{m} \sum_{\tau \in \mathcal{T}} M_T^{(\tau)}, \qquad \hat{y}(i) = \arg\max_c \bar{M}_T^{(c)}(i).$$

Small $\tau$ downweights cross-edge transport; large $\tau$ encourages within-region smoothing. The average $\bar{M}_T$ behaves like a multi-bandwidth anisotropic diffusion, improving robustness across appearances.

**Time Complexity.** Let $N = HW$, classes $C+1$, directions $P = 8|D|$, iterations $T$, and $m = |\mathcal{T}|$ (for NAMR). Constructing $\bar{r}$ and $\sigma$ costs $\mathcal{O}(N \cdot 3 \cdot P)$. Each message-passing step costs $\mathcal{O}(N \cdot C \cdot P)$. Hence:

$$\text{PAMR: } \mathcal{O}\big(N(3P + TCP)\big), \qquad \text{NAMR: } \mathcal{O}\big(N(3P + mTCP)\big).$$

The framework is drop-in: RGB can be replaced by any guidance features (e.g., depth encoders) without changing derivations.

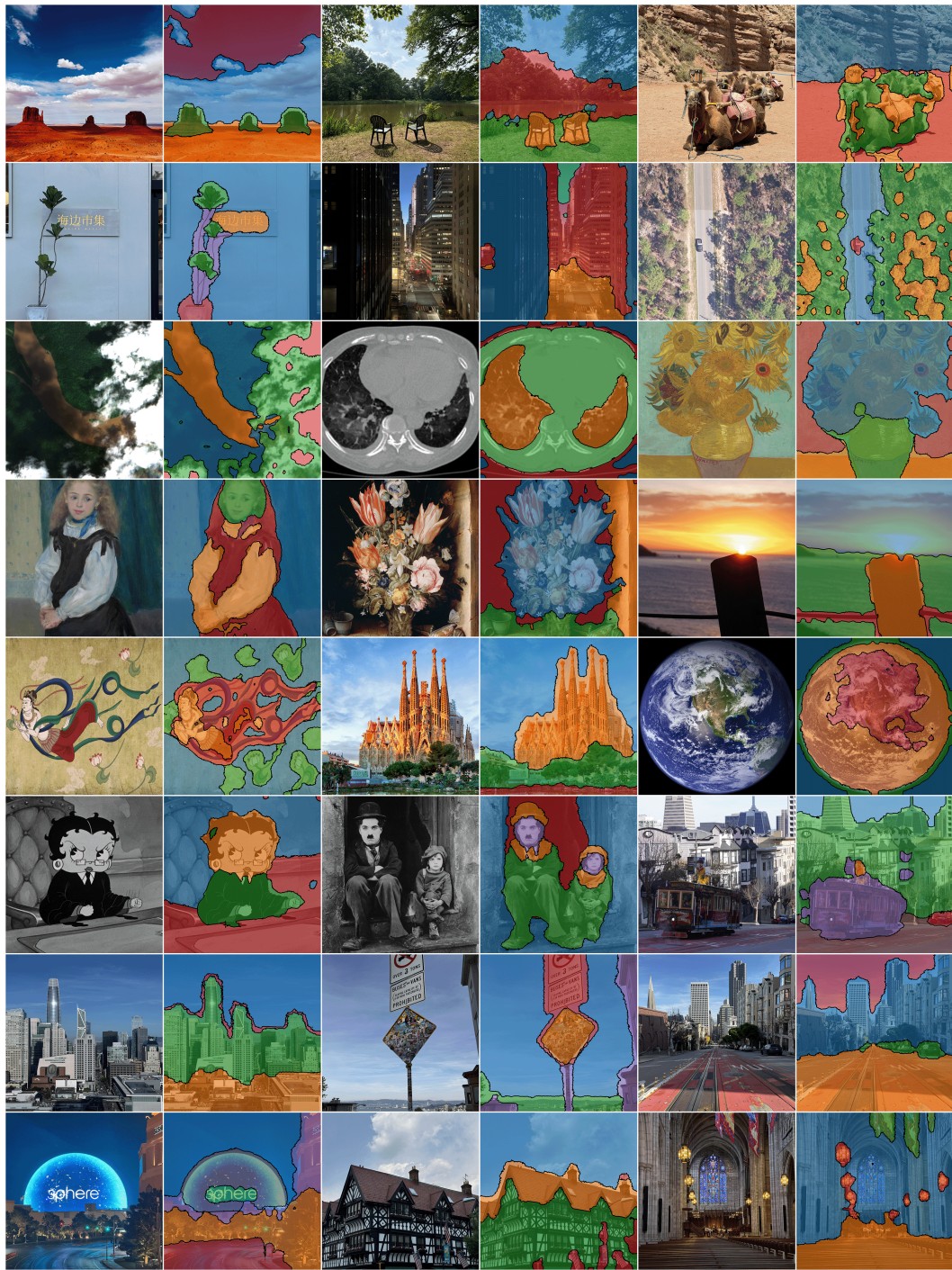

Figure 5: **Qualitative segmentation results of Falcon across diverse domains.** Examples span natural scenes, urban landscapes, medical CT, paintings, cartoons, and planetary imagery. Falcon consistently produces coherent partitions with sharp boundaries and stable region assignment, demonstrating strong generalization beyond a single modality.

## F  FALCON SEGMENTATION RESULTS VISUALIZATION

## G   END-TO-END RUNTIME ACROSS DATASETS AND ENCODERS

Table 3 reports the *end-to-end* evaluation time across all dataset–encoder pairs, decomposed into *Data Preparation*, *Feature Extraction*, and *Mask Generation*. Unlike Appendix I, which isolates graph-cut time and refinement time under controlled post-processing settings, this table summarizes the full pipeline runtime used in the main paper.

Falcon achieves the lowest total runtime across all 18 dataset–encoder pairs. Representative examples include: (i) *DINOv3-B, Cityscapes*: Total Time drops from 784.04s (DiffCut) to 87.47s ($\sim$8.9$\times$), and Mask Generation from 747.97s to 52.49s ($\sim$14.3$\times$); (ii) *SD2.1, Context*: Total Time reduces from 2535.03s to 1173.87s ($\sim$2.2$\times$), and Mask Generation from 1546.75s to 182.86s ($\sim$8.5$\times$); (iii) *SSD-1B, Cityscapes*: even when preprocessing dominates, Falcon still lowers Total Time from 177.27s to 114.67s ($\sim$1.5$\times$) and reduces Mask Generation from 78.11s to 15.46s ($\sim$5.0$\times$).

Table 3: **End-to-end evaluation time across datasets and encoders.** Best **Total Time** (s) and **Mask Generation** (s) for each dataset–encoder pair are bold.

| Encoder | Benchmarks | Method | Total Time | Data Preparation | Feature Ext. | Mask Generation |
|---|---|---|---|---|---|---|
| **SSD-1B** | Cityscapes | AutoSC | 126.63 | 23.02 | 76.25 | 27.36 |
| | | DiffCut | 177.27 | 22.97 | 76.19 | 78.11 |
| | | Falcon | **114.67** | 22.96 | 76.25 | **15.46** |
| | VOC | AutoSC | 368.96 | 68.61 | 220.71 | 79.64 |
| | | DiffCut | 483.05 | 71.21 | 220.39 | 191.45 |
| | | Falcon | **335.31** | 68.88 | 220.96 | **45.47** |
| | Context | AutoSC | 1288.12 | 232.30 | 775.63 | 280.19 |
| | | DiffCut | 1700.30 | 237.37 | 775.37 | 687.56 |
| | | Falcon | **1170.80** | 233.73 | 775.74 | **161.33** |
| | COCO-Stuff | AutoSC | 551.18 | 100.89 | 330.78 | 119.51 |
| | | DiffCut | 734.07 | 99.92 | 330.73 | 303.42 |
| | | Falcon | **497.55** | 98.84 | 330.91 | **67.80** |
| | COCO-Object | AutoSC | 556.11 | 105.84 | 330.72 | 119.55 |
| | | DiffCut | 748.96 | 114.85 | 330.68 | 303.43 |
| | | Falcon | **506.54** | 106.48 | 331.23 | **68.83** |
| | ADE20K | AutoSC | 505.73 | 91.34 | 304.17 | 110.22 |
| | | DiffCut | 686.08 | 90.65 | 304.03 | 291.40 |
| | | Falcon | **459.25** | 92.13 | 304.77 | **62.35** |
| **SD2.1** | Cityscapes | AutoSC | 129.15 | 23.13 | 74.44 | 31.58 |
| | | DiffCut | 410.24 | 23.12 | 74.27 | 312.85 |
| | | Falcon | **114.35** | 23.13 | 74.58 | **16.64** |
| | VOC | AutoSC | 374.69 | 68.68 | 214.97 | 91.04 |
| | | DiffCut | 719.16 | 68.37 | 214.77 | 436.02 |
| | | Falcon | **335.38** | 69.51 | 214.62 | **51.25** |
| | Context | AutoSC | 1313.00 | 234.68 | 756.28 | 322.04 |
| | | DiffCut | 2535.03 | 233.09 | 755.19 | 1546.75 |
| | | Falcon | **1173.87** | 234.24 | 756.77 | **182.86** |
| | COCO-Stuff | AutoSC | 561.70 | 101.46 | 322.60 | 137.64 |
| | | DiffCut | 1197.13 | 99.31 | 322.27 | 775.55 |
| | | Falcon | **498.58** | 97.88 | 322.90 | **77.80** |
| | COCO-Object | AutoSC | 566.79 | 106.95 | 322.52 | 137.32 |
| | | DiffCut | 1208.45 | 103.69 | 322.18 | 782.58 |
| | | Falcon | **503.54** | 102.28 | 323.37 | **77.89** |
| | ADE20K | AutoSC | 514.99 | 91.26 | 296.62 | 127.11 |
| | | DiffCut | 1090.35 | 90.80 | 296.25 | 703.30 |
| | | Falcon | **457.40** | 89.28 | 296.60 | **71.52** |
| **DINOv3-b** | Cityscapes | AutoSC | 242.67 | 23.07 | 12.14 | 207.46 |
| | | DiffCut | 784.04 | 23.93 | 12.14 | 747.97 |
| | | Falcon | **87.47** | 22.48 | 12.50 | **52.49** |
| | VOC | AutoSC | 718.39 | 69.10 | 35.00 | 614.29 |
| | | DiffCut | 1950.92 | 66.33 | 34.97 | 1849.62 |
| | | Falcon | **238.46** | 66.34 | 35.12 | **137.00** |
| | Context | AutoSC | 2528.36 | 236.62 | 122.93 | 2168.81 |
| | | DiffCut | 6935.53 | 234.54 | 122.85 | 6578.14 |
| | | Falcon | **903.28** | 237.21 | 123.24 | **542.83** |
| | COCO-Stuff | AutoSC | 1073.20 | 100.08 | 52.47 | 920.65 |
| | | DiffCut | 2916.03 | 99.05 | 52.43 | 2764.55 |
| | | Falcon | **384.14** | 101.16 | 52.41 | **230.57** |
| | COCO-Object | AutoSC | 1184.79 | 107.20 | 52.47 | 1025.12 |
| | | DiffCut | 2947.12 | 102.48 | 52.43 | 2792.21 |
| | | Falcon | **386.91** | 102.80 | 52.72 | **231.39** |
| | ADE20K | AutoSC | 996.57 | 93.55 | 48.26 | 854.76 |
| | | DiffCut | 2787.44 | 90.47 | 48.22 | 2648.75 |
| | | Falcon | **351.39** | 90.10 | 48.46 | **212.83** |

Table 4: **Controlled comparison with MaskCut under matched settings (mIoU).** All methods are evaluated under the same class-agnostic semantic segmentation protocol. "Reported" denotes numbers taken from the DiffCut paper; "our reported" denotes re-implementation in our unified codebase.

| Method | Encoder | VOC | Context | COCO-Object | COCO-Stuff-27 | Cityscapes | ADE20K |
|---|---|---|---|---|---|---|---|
| MaskCut $k=5$ (reported by DiffCut) | Unknown | 53.80 | 43.40 | 30.10 | 41.70 | 18.70 | 35.70 |
| MaskCut $k=5$ (our reported) | DINOv3-B Key | **81.81** | 30.46 | 51.97 | 21.70 | 2.74 | 14.83 |
| MaskCut $k=5$ (our reported) | DINOv3-B Token | 73.67 | 33.26 | 52.87 | 25.54 | 6.06 | 17.48 |
| Falcon + CRF | DINOv3-B Token | 75.15 | **44.54** | **62.06** | **30.71** | **25.40** | **42.04** |

## H  CONTROLLED COMPARISONS WITH MASKCUT AND DENSECRF

To complement the main-paper comparison against published baselines, we additionally perform controlled comparisons with MaskCut under matched settings. The goal is to isolate the contribution of the segmentation solver itself by aligning the encoder, resolution, evaluation protocol, and post-processing choices.

Specifically, we re-implement MaskCut within our unified codebase and evaluate it using the same image resolution, the same class-agnostic Hungarian-matching protocol, and matched post-processing settings for controlled comparison. Since the original MaskCut paper uses ViT key embeddings together with DenseCRF, we report both key-feature and token-feature variants of Mask-Cut. We also evaluate Falcon with DenseCRF to provide a direct comparison under the same post-processing family.

The results show that Falcon consistently outperforms the controlled MaskCut variants even when using the same token features and the same DenseCRF post-processing. This confirms that the gains of Falcon come from directly optimizing the discrete $K$-way NCut objective, rather than from encoder or refiner differences alone.

Under matched token features and the same DenseCRF post-processing, Falcon clearly outperforms MaskCut across all six benchmarks. This controlled comparison strengthens the main-paper results by showing that Falcon's gains are attributable to the discrete $K$-way solver itself, rather than to backbone choice or post-processing differences.

## I  RUNTIME DECOMPOSITION BY GRAPH CUT AND REFINEMENT

The runtime analysis in the main paper focuses on end-to-end evaluation time. To further clarify where the speedup comes from, we decompose inference into *graph-cut time* and *refinement time*. This decomposition is particularly important for comparing Falcon with recursive NCut methods and with different post-processing modules.

All methods are measured under the same hardware and implementation setting. In particular, we use the same image resolution, the same encoder setting, and the same evaluation protocol, and we report wall-clock time for the graph-partitioning stage and the subsequent refinement stage separately. For MaskCut, we pair the recursive NCut solver with DenseCRF, following its classical refinement setup. For Falcon, we report three post-processing choices: PAMR, NAMR, and Dense-CRF.

The results show that Falcon's efficiency gains come primarily from the graph-partitioning solver itself. Even when paired with the same DenseCRF post-processing, Falcon remains substantially faster than MaskCut across all six benchmarks. Moreover, PAMR and NAMR contribute only a small overhead relative to Falcon's graph-cut stage, whereas DenseCRF introduces a larger but still secondary refinement cost.

Across all six benchmarks, Falcon remains substantially faster than MaskCut even when both are paired with DenseCRF. For example, on Pascal Context, Falcon+CRF reduces graph-cut time from 2537.03s to 542.73s, while also reducing refinement time from 310.40s to 217.63s. On Cityscapes, Falcon+CRF reduces graph-cut time from 241.28s to 52.37s. These comparisons confirm that the dominant speedup comes from replacing recursive spectral partitioning with Falcon's direct discrete $K$-way solver.

Table 5: **Runtime decomposition into graph-cut time and refinement time (seconds).** All numbers are measured under the same codebase and hardware setting.

| Method | VOC | | Context | | COCO-Object | | COCO-Stuff-27 | | Cityscapes | | ADE20K | |
|---|---|---|---|---|---|---|---|---|---|---|---|---|
| | GraphCut | Refine | GraphCut | Refine | GraphCut | Refine | GraphCut | Refine | GraphCut | Refine | GraphCut | Refine |
| MaskCut (with CRF) | 720.75 | 80.18 | 2537.03 | 310.40 | 1078.50 | 125.21 | 1077.81 | 149.24 | 241.28 | 32.59 | 1001.57 | 120.20 |
| Falcon (with PAMR) | 137.00 | 4.37 | 542.83 | 15.77 | 231.39 | 6.61 | 230.57 | 7.18 | 52.49 | 1.70 | 212.83 | 6.57 |
| Falcon (with NAMR) | 136.77 | 12.88 | 541.50 | 44.65 | 231.56 | 21.05 | 231.20 | 23.40 | 52.39 | 5.02 | 211.34 | 20.55 |
| Falcon (with CRF) | 137.73 | 58.74 | 542.73 | 217.63 | 230.69 | 108.46 | 230.55 | 115.09 | 52.37 | 27.92 | 212.78 | 97.86 |

Table 6: **Peak GPU memory usage (GB) at different input resolutions.** Numbers are averaged over six datasets using DINOv3-B with batch size 1 on an RTX 6000 Ada GPU.

| Resolution | MaskCut | DiffCut | Falcon |
|---|---|---|---|
| $256^2$ | 0.51 | 0.51 | 0.51 |
| $512^2$ | 0.54 | 0.54 | 0.54 |
| $768^2$ | 0.59 | 0.82 | 0.59 |
| $1024^2$ | 1.00 | 1.95 | 0.93 |
| $2048^2$ | 10.52 | 23.71 | 9.48 |

The table also shows that PAMR and NAMR add only a modest overhead relative to Falcon's graph-cut stage. For instance, on VOC, Falcon requires 137.00s for graph cut and only 4.37s for PAMR refinement; even NAMR remains comparatively lightweight at 12.88s. DenseCRF is more expensive than PAMR or NAMR, but the solver-level advantage of Falcon persists under the same CRF post-processing family.

Overall, these results support the main-paper runtime analysis from a complementary angle: Falcon is not merely faster because of a lighter refiner, but because its graph-cut stage is substantially more efficient than recursive NCut baselines.

## J    MEMORY USAGE AT DIFFERENT INPUT RESOLUTIONS

Since dense affinity methods scale quadratically in the number of tokens, memory usage is an important practical consideration in addition to runtime. To quantify this aspect, we measure peak GPU memory at different input resolutions for MaskCut, DiffCut, and Falcon on top of DINOv3-B. All measurements are averaged over the six benchmark datasets used in the main paper, with batch size 1 on an NVIDIA RTX 6000 Ada GPU.

The results show that Falcon is substantially more memory-efficient than DiffCut at high resolution, while remaining comparable to or slightly better than MaskCut. In particular, at $2048 \times 2048$, Falcon uses about $2.5\times$ less peak GPU memory than DiffCut (9.48GB vs. 23.71GB), supporting its practicality for larger token graphs.

At lower resolutions, all three methods have similar memory footprints, as the token graph remains relatively small. As the input resolution increases, however, the gap becomes more pronounced: DiffCut incurs the largest memory growth, whereas Falcon remains much closer to MaskCut while still optimizing a global discrete $K$-way objective. This complements the runtime analysis in the main paper and shows that Falcon is not only faster in practice, but also more scalable in memory than prior recursive NCut-based methods at high resolution.

## K    SENSITIVITY TO THE INERTIA AND BACKTRACKING PARAMETERS

Falcon uses a proximal inertia parameter together with a monotone backtracking rule. To examine the robustness of these optimization parameters, we sweep the initial inertia weight $\tau_0$ and the backtracking growth factor $\gamma$ on Cityscapes using two representative encoders, DINOv3-B16 and SSD-1B. Unless otherwise noted, Falcon uses the same default values $\tau_0 = 1$ and $\gamma = 2$ across datasets and pretrainings.

The results show that Falcon is highly insensitive to these settings: the final mIoU varies by less than 0.1% across the tested configurations. Under the default setting, Falcon typically converges in essentially one outer iteration per image with negligible backtracking. Even under more aggressive

Table 7: **Sensitivity to the inertia and backtracking parameters on Cityscapes.** mIoU remains essentially unchanged across settings, while the default configuration converges in about one outer iteration per image.

| Encoder | $\tau_0$ | $\gamma$ | Outer iters/img | Backtrack rate | mIoU (%) |
|---|---|---|---|---|---|
| DINOv3-B16 | 0.50 | 1.50 | 1.22 | 43.30 | 25.27 |
| DINOv3-B16 | 0.50 | 5.00 | 1.22 | 43.40 | 25.27 |
| DINOv3-B16 | 1.00 | 2.00 | 1.00 | 0.00 | 25.29 |
| SSD-1B | 0.50 | 1.50 | 1.44 | 67.00 | 30.56 |
| SSD-1B | 0.50 | 5.00 | 1.21 | 67.80 | 30.56 |
| SSD-1B | 1.00 | 2.00 | 1.02 | 1.20 | 30.56 |

initializations, backtracking acts as a safety mechanism and increases runtime only slightly, without changing the final segmentation quality.

Under the standard choice $(\tau_0, \gamma) = (1.0, 2.0)$, Falcon converges in essentially one outer iteration with almost zero backtracking. This supports the practical runtime observations in the main paper and shows that the backtracking rule functions mainly as a stability safeguard rather than as a sensitive tuning knob.

In all reported experiments, we set a small upper bound $T_{\max} = 5$ and stop early when the objective converges. In practice, the stopping condition is typically met after the first outer iteration. Likewise, the line search never behaves like a heavy inner loop: under the default setting, the descent condition is usually satisfied on the first trial, and even with a smaller initialization such as $\tau_0 = 0.5$, the additional overhead remains negligible relative to one graph-cut update.

