# OpenReview forum: "Falcon: Fast Proximal Linearization of Normalized Cuts for Unsupervised Image Segmentation"
_ICLR.cc/2026/Conference — ICLR 2026 Poster_

### Official Review · Reviewer_mrYD · 2025-10-27

**Soundness:** 3
**Presentation:** 2
**Contribution:** 3
**Rating:** 6
**Confidence:** 2

**Summary:**

This paper proposes “Falcon,” a graph cut algorithm for performing unsupervised image segmentation using embeddings from pretrained vision transformers. This algorithm is a proximal-gradient solver that directly optimizes the discrete K-way NCut objective without spectral relaxation. The paper shows this proposed method outperforms baseline approaches on common datasets and provides ablation and runtime experiments.

**Strengths:**

Technical novelty and practical impact
- The authors propose a new graph cut method that is both performant and efficient for solving unsupervised segmentation problems provided a pretrained vision transformer.
- The authors show improved performance compared to baseline methods (MaskCLIP, MaskCut, DiffSeg, DiffCut, AutoSC) on common benchmark datasets (VOC, Context, COCO-Object, COCO-Stuff-27, Cityscapes, ADE20K), suggesting the proposed method improves unsupervised segmentation with pretrained transformer models.

Helpful ablation experiments
- The authors perform helpful ablations showing the method improves performance over multiple pretrained networks and evaluating the refiner on both the proposed method as well as with baseline methods.
- The authors provide detailed runtime experiments showing the method improves inference speed compared to two baselines.

**Weaknesses:**

Baseline method selection
- The authors focus on a few previously published methods and show the proposed method convincingly outperforms them, but do not convince the reader these are representative/comprehensive baselines. Authors should clarify how and why they choose what baseline to include vs. exclude (e.g., zero shot SAM; STEGO; diffuse attend and segment; U2Seg; etc.), and consider including additional baselines if there is no convincing reason they should be excluded.

Reproducibility
- Will the authors publish code to make this method usable/reproducible? If not, I have doubts about the reproducibility of this method.

There are a few items that need to be clarified to contextualize the strengths of the method:
- Why are images resized to 1024 x 1024? This seems higher resolution than is typical for these natural image datasets and would increase runtime with the pretrained transformers. It also seems like this resize may be inflating inference speedups.
- How is K determined? You say “K is determined via spectral thresholding: we count eigenmodes below a negative threshold κ and clamp K to a predefined range, “ but it is unclear to me what this means. Is K a range, and final K value is selected via best empirical performance? Does this need to be hand tuned per dataset? Where in the method pipeline is this operation performed? What is the rationale for the negative threshold and clamping? Is kappa hand tuned per dataset?
- Can the authors comment on how memory usage compares to baseline methods? Since many of the intermediate matrices are NxN, is there a limit on image/token size * feature size (both dependent on the pretrained network) that is practical with this method?
- Since gradient scores are weighted by cluster volumes, how do the authors expect volume size to impact performance? Is this dependence seen in the empirical results? (e.g., significantly stronger performance on large objects vs. small)

Minor weaknesses:
- I think the authors have used \cite{}, which includes citations in-text, where they should have used \citep{}, which includes citations in parentheses. The authors should switch the citations to the correct format as appropriate to improve readability.
- Typo on line 43, “as” instead of “has.”
- The authors lay out 3 challenges with existing segmentation methods in the abstract. Similarly, they list 3 challenges with existing segmentation methods in the introduction. However, these 3 challenges are not the same in the abstract vs. intro. This makes it challenging as a reader to know what limitations with previous methods the authors are aiming to address. The authors should make the abstract and intro consistent with one another.
- Line 242, “(iii)” is repeated twice.
- The authors lay out issue with “classical refiners” in lines 295-297, but do not state how PAMR or NAMR get around these challenges. A transition paragraph would help.
- (H, W) seems to be overloaded notation. The authors use (H,W) to describe the size of the intermediate mask/features in Section 3.3, but then use (H,W) to describe the full resolution image in Section 3.4. Are (H,W) always the same, i.e. is the intermediate mask in Section 3.3 always at full resolution? If so, that is not clear from Section 3.3, as you refer to this stage as “intermediate-resolution refinement.”
- There are no segmentation visualizations in the main text, they are instead put in the supplementary. If the authors have space, consider moving these to the main text. Additionally consider including visualizations of baseline vs. Falcon results for the reader to qualitatively understand differences in the methods.
- There are many parameters given in Section 3; authors should include defaults for each of these in either the main text or supplementary.

**Questions:**

Addressed in Weaknesses section.

---

> ### Author Response · Authors · 2025-11-26
> **Response to Reviewer mrYD (Part I)**
>
> We thank Reviewer mrYD for the positive feedback and for recognizing the novelty of our method. We address the specific questions below.
>
> ## *W1: Baseline method selection.*
>
> Our work is scoped to the **training-free setting**, where a frozen pretrained encoder conducts segmentation at only inference time, without any additional training. For this reason, our main comparisons focus on methods that are also training-free, namely MaskCut, AutoSC, DiffSeg, and DiffCut. Regarding the specific methods mentioned by the reviewer, we would like to clarify the following:
> - **SAM:** A supervised foundation segmentation model trained on one billion annotated masks.
> - **STEGO and U2Seg:** Human annotation-free (unsupervised) approaches that rely on self-training.
> - **Diffuse, attend, and segment.** This is represented in our baselines by **DiffSeg**, which we already include in Table 1 of the submission.
>
> We provide a refined table that aggregates reported results from the original papers of these methods alongside the training-free methods we compare against. We annotate each method with its characteristics such as AM (Annotated Mask) and AT (Adaptive Training).
>
>
> | Method | AM | AT |  VOC | Context | COCO-Object | COCO-Stuff-27 | Cityscapes | ADE20K |
> | :--- | :---: | :---: | :---: | :---: | :---: | :---: | :---: | :---: |
> | SAM3 (Category prompt) | ✓ |  ✓ | - | 60.80 | - | - | 65.20 | 39.00 |
> |  |  |  |  |  |  |  |  |  |
> | U2Seg | ✗ |  ✓ | - | - | - | 30.20 | - | - |
> | STEGO | ✗ |  ✓ | - | - | - | 28.20 | 21.00 | - |
> |  |  |  |  |  |  |  |  |  |
> | MaskCLIP | ✗ |  ✗ | 38.80 | 23.60 | 20.60 | 19.60 | 10.00 | 9.80 |
> | MaskCut | ✗ | ✗ | 53.80 | 43.40 | 30.10 | 41.70 | 18.70 | 35.70 |
> | DiffSeg | ✗ | ✗ | 49.80 | 48.80 | 23.20 | 44.20 | 16.80 | 37.70 |
> | DiffCut | ✗ | ✗ | 65.20 | 56.50 | 34.10 | 49.10 | 30.60 | 44.30 |
> |  |  |  |  |  |  |  |  |  |
> | AutoSC† (ours, refined) | ✗ | ✗ | 77.57 | 57.27 | 61.56 | 49.39 | 25.72 | 40.10 |
> | DiffCut† (ours, refined) | ✗ | ✗ | 71.68 | **58.17** | 61.65 | 49.18 | 30.77 | 44.40 |
> |  |  |  |  |  |  |  |  |  |
> | **Falcon (ours)** | ✗ | ✗ | **78.40** | 57.15 | **61.80** | **50.37** | **33.69** | **45.17** |
> |  |  |  |  |  |  |  |  |  |
>
>
> We will clarify this scope and add the aforementioned summary table in the revision so that readers can better see (i) where Falcon sits in the broader unsupervised segmentation landscape, and (ii) that our chosen baselines are representative within the training-free setting we target.
>
>
> ## *W2: Reproducibility and code release.*
>
> We fully agree that reproducibility is crucial. Upon acceptance, we will release the full codebase for Falcon together with setup and run scripts that configure the environment and reproduce experiments with one command.
>
>
> ## *W3: Why are images resized to $1024 \times 1024$?*
>
> Choosing $1024 \times 1024$ follows the protocol of the previous training-free **SOTA** DiffCut, which apply SSD-1B encoder at this resolution.
>
> We adopt the same setting to ensure a fair, apples-to-apples comparison. In the final version, we will additionally report results at several lower resolutions.
>
>
>
> ## *W4: How is \(K\) determined via spectral thresholding?*
>
> We determine $K$ using a standard spectral heuristic: we compute eigenvalues of the normalized affinity matrix, and count those below a cutoff $\kappa$ as the number of segments $K$. This step is computationally negligible compared to feature extraction and the following proximal iterations. As shown in Fig. 4 (right) in the submission, varying $\kappa$ within a broad, semantically reasonable interval directly changes the estimated $K$, but leaves the final segmentation quality (mIoU) relatively stable on both VOC and Cityscapes. Hence, these results indicate that using $\kappa$ within a reasonable range is robust in terms of performance.
>
> In practice, we fix a single $\kappa$ per dataset rather than tuning it per image. Empirically, we find a consistent pattern: datasets with sparse annotations in iconic scenes (a few large objects) such as Pascal VOC work well with a cutoff around $\kappa=0.0$, whereas more densely annotated, scene-centric datasets (Cityscapes, COCO-Stuff, COCO-Object, Context, ADE20K) with more small objects share a higher threshold around $\kappa=0.3$. This dataset-specific choice is natural, because each benchmark defines a different segmentation task and annotation protocol, so there is no single universally appropriate granularity across all datasets. As acknowledged in the segmentation literature, from classical NCut to modern models such as SAM that support multi-granularity masks, the correct number of segments is inherently ambiguous and depends on how fine or coarse the annotation standard is. We will update this discussion in the revised version.

---

> ### Author Response · Authors · 2025-11-26
> **Response to Reviewer mrYD (Part II)**
>
> ## *W5: Memory usage and practical limits.*
>
> We agree that memory usage is an important aspect. As the reviewer notes, any method that operates on **dense token affinities** incurs \(\mathcal{O}(N^2)\) memory in the number of tokens \(N\). Falcon does not change this scaling, but it removes recursive eigen-decompositions and repeated NCut passes, which substantially reduces the constants.
>
> To quantify this, we measured the average peak GPU memory (GB) of MaskCut, DiffCut, and Falcon at different input resolutions on top of DINOv3-B, averaged over six datasets, with batch size 1 on an RTX 6000 Ada. The results (reported in the revised appendix) show that:
>
> | Resolution | MaskCut            | DiffCut            | **Falcon (ours)**   |
> |--------------------------:|--------------------|--------------------|---------------------|
> | $256^2$                   | **0.5112 ± 0.0000**    | **0.5112 ± 0.0000**    | **0.5112 ± 0.0000** |
> | $512^2$                   | **0.5362 ± 0.0018**    | 0.5374 ± 0.0018    | **0.5362 ± 0.0018** |
> | $768^2$                   | **0.5927 ± 0.0000**    | 0.8201 ± 0.0350    | **0.5927 ± 0.0000** |
> | $1024^2$                  | 1.0013 ± 0.0000    | 1.9510 ± 0.2622    | **0.9269 ± 0.0000** |
> | $2048^2$                  | 10.5234 ± 0.0000   | 23.7140 ± 3.5698   | **9.4759 ± 0.0000** |
>
> As shown, Falcon scales much more gracefully than DiffCut. Notably, at $2048 \times 2048$, Falcon is 2.5x more memory-efficient than DiffCut (9.48GB vs. 23.71GB) and even slightly more efficient than the simpler MaskCut baseline. This confirms that Falcon is well-suited for high-resolution scenarios compared to the previous methods. We will add this memory analysis and a short discussion in the revised version.
>
>
> ## *W6: Effect of volume terms on large vs. small objects.*
>
> The concern behind "gradient scores weighted by cluster volumes" is whether Falcon might implicitly favor large objects. We clarify two key points:
>
> - **No bias toward specific sizes of objects.** The volume terms do not create a bias toward large or small objects. They act as a normalization that prevents trivial solutions (e.g., assigning all pixels to a single cluster), in line with the standard NCut formulation.
>
> - **The volume terms are mathematically derived, not hand-crafted.** The volume terms are not a heuristic designed to favor specific object sizes. They emerge directly from taking the gradient of the discrete NCut objective. Therefore, the gradient simply follows the direction of steepest descent for the NCut objective.
>
> Formally, the discrete NCut objective is a sum of ratios:
>
> $$
> f(X) = \sum_k \frac{q_k}{v_k}, \quad
> q_k = X_{\cdot k}^\top W X_{\cdot k}, \;
> v_k = X_{\cdot k}^\top d,
> $$
>
> where each cluster’s association $q_k$ is **normalized by its own volume** $v_k$. Differentiating this objective yields update scores of the form :
>
> $$
> \nabla_{X}f \propto \frac{(WX) _ {ik}}{v_k}-\frac{q_k}{v_k^2} d_i X _ {iK} ,
> $$
>
> so the dependence on cluster size appears through the **inverse factors** $1/v_k$ and $1/v_k^2$. In other words, larger clusters are actually *down-weighted* by these factors and do not dominate the updates simply because they contain more pixels. This structure is a direct mathematical consequence of optimizing the volume-normalized NCut objective.
>
> Empirically, we do not observe any skew toward large objects. On datasets with strong scale variation, Falcon consistently improves over strong baselines: on Cityscapes (where large "road" regions coexist with thin "pole" and "traffic light" categories), Falcon improves over DiffCut by +3.1 mIoU, and on Pascal VOC (iconic, large objects) and COCO-Stuff-27 (complex, scene-centric layouts), it achieves gains of +13.2 mIoU and +1.3 mIoU over DiffCut, respectively. If the volume terms were biased toward large regions, we would expect performance degradation on scene-centric benchmarks such as Cityscapes or COCO-Stuff, which we do not observe. Overall, these results suggest that the volume terms stabilize the optimization and debias NCut across clusters, rather than sacrificing small-object performance.

---

> ### Author Response · Authors · 2025-11-26
> **Response to Reviewer mrYD (Part III)**
>
> ## *Minor weaknesses*
>
> We thank the reviewer for the detailed comments on presentation aspects and will address all of them in the revision.
>
> - We will fix the citation style (using parenthetical citations where appropriate) and correct the minor typos noted by the reviewer.
>
> - For the "three challenges" described in the abstract and the introduction, we will unify them into a single, consistent set and ensure that the wording is aligned.
>
> - For the discussion of classical refiners vs. PAMR/NAMR, we will add timing comparisons and discussion between DenseCRF and PAMR/NAMR in the revised version. To clarify, our intention is not to present PAMR or NAMR as core contributions, but to use them as fair and modern refiners for comparison.
>
> - For the overloaded $(H, W)$ notation, we will clean up the notation and adjust the wording around “intermediate-resolution refinement” to clearly indicate the resolutions used in our implementation.
>
> - For qualitative visualizations, although we initially placed all segmentation examples in the appendix due to space constraints, we will move a subset of representative visualizations to the main paper in the revised version.
>
> - Finally, we will provide a more systematic description of all key hyperparameters used in our experiments, including their roles and default values, in the revised version.

---

### Official Review · Reviewer_uM5A · 2025-10-29

**Soundness:** 2
**Presentation:** 2
**Contribution:** 3
**Rating:** 2
**Confidence:** 3

**Summary:**

This paper presents a method for unsupervised semantic image segmentation. The begins by image feature extraction using pre-trained models such as Vision Transformer trained with DINO. An affinity matrix is assembled by pairwise feature similarity. The affinity matrix elements are viewed as weights on graph edges. This enables a bipartitioning objective, where the resulting partitions equate to segments in the input image. This paper presents Falcon, an extension of Normalized Cuts and recently proposed Mask Cut algorithms that partition the graph in a recursive fashion; more than $K=2$ partitions are obtained by recursively partitioning the remaining subgraph. Instead, Falcon partitions into $K>2$ groups at once and avoids eigenvalue decomposition, reducing the method complexity from $\mathcal{O}(N^3)$ to $\mathcal{O}(KN^2)$. Since tokens are subsampled wrt. the input image resolution, the authors propose two methods to recover partitioning at a higher resolution.

**Strengths:**

This paper considers an important topic in computer vision. Contributions to semantic image segmentation are meaningful and relevant. The work is also well motivated; current methods based on graph partitioning depend on eigenvalue decomposition that is $\mathcal{O}(N^3)$ in time, making them unscalable as input resolution grows. The experimental section includes evaluations on six datasets.

**Weaknesses:**

The method section is really complex and difficult to follow. There is an abundance of symbols that are not gradually included, and the text between the equations does not connect the formalism in understandable fashion. For example, most quantities are not expressed in text: readers would appreciate understanding what do $\mathbf{v}$ and $\mathbf{q}$ hold, and that $v_k$ and $c_k$ are merely elements within these vectors. Some variables are multiplied. For instance, a square similarity matrix $\mathbf{S}$ is introduced in Equation (2). Afterwards, a $N \times K$ matrix $\mathbf{S}$ is used in Equation (8). Similarly, $c_k$ (Equation(9)) and $\mathbf{c}_k$ (Equation(13)) refer to different quantities. Equation (11) expresses the update in the proposed iterative algorithm, but it seems to render improperly in the PDF.

The experimental section lacks proper comparison with related work.
Firstly, influential relevant related methods, U2Seg [1] and UnSAM [2], are not included in experimental comparison.
Secondly, since CutLer uses MaskCut to generate instance segmentation pseudo labels, it is unclear how is Mask Cut adapted for unsupervised semantic segmentation. This is important to note as Falcon is most related to Mask Cut. Moreover, the original work uses DINOv1 and CRF [3] to generate instance pseudo labels. A fair comparison should include the same backbone and postprocessing to highlight the contribution. Furthermore, the proposed postprocessing methods are not validated against the commonly used CRF postprocessing, therefore there is no validation of the proposed postprocessing methods.

Execution time measurements do not include a comparison with Mask Cut. Also, separate time measurements for graph partitioning and postprocessing (the two contributions) are missing.
The authors comment that CRF post processing is CPU-bound. This is true for the official Mask Cut implementation [4], but does hold in general [5]. To highlight the lower theoretical complexity of Falcon, the experimental section could include measurements under common implementation and hardware*.

- [1] Niu, Dantong, et al. "Unsupervised universal image segmentation." Proceedings of the IEEE/CVF conference on computer vision and pattern recognition. 2024.
- [2] Wang, XuDong, Jingfeng Yang, and Trevor Darrell. "Segment anything without supervision." Advances in Neural Information Processing Systems 37 (2024): 138731-138755.
- [3] Krähenbühl, Philipp, and Vladlen Koltun. "Efficient inference in fully connected crfs with gaussian edge potentials." Advances in neural information processing systems 24 (2011).
- [4] https://github.com/facebookresearch/CutLER/blob/main/maskcut/maskcut.py
- [5] https://github.com/heiwang1997/DenseCRF

$*$ Note that Mask Cut performs eigenvalue decomposition on CPU.

**Questions:**

- The number of clusters $K$ is "is determined via spectral thresholding". What is the complexity of this algorithm?
- How does the accuracy depend on the number of iterations $\mathbf{X}^{(t)}$? How many iterations does Falcon use in practice?
- How many iterations are required to perform the line search in each iteration from Equation (12)?

---

> ### Author Response · Authors · 2025-11-27
> **Response to Reviewer uM5A (Part I)**
>
> We thank Reviewer uM5A for the careful and constructive review, which helps us clarify the method and strengthen the presentation.
>
>
>
> ## *W1: The method description and notation.*
> We appreciate the reviewer’s detailed comments on the presentation of Section 3. We would like to clarify that the issues raised are notational and do not affect the underlying algorithm.
>
> In Sec. 3.2 we denote by $q_k = x_k^\top W x_k$ the intra-cluster association and by $v_k = x_k^\top D x_k$ the cluster volume (sum of node degrees in cluster $k$). The vectors $q$ and $v$ simply collect these quantities across clusters, and $q_k$, $v_k$ are their entries. In Eq. (9), the scalar $c_k = q_k / v_k^2$ is a shorthand that appears in the gradient expression. In Sec. 3.3, the symbol $c_k$ is the high-resolution feature prototype of cluster $k$. For overloading of $S$ and $c_k$ (similarity matrix vs. cached scores; scalar coefficient vs. feature prototype), we will rename these quantities and refine the rendering of Eq. (11) in the revised version so that the update step is clearly readable.
>
>
> ## *W2: Proper comparison with related work and MaskCut/CRF baselines.*
>
>
> ### Why not compare with U2Seg and UnSAM?
> Regarding U2Seg and UnSAM, they both require training. In contrast, our work is explicitly scoped to the training-free setting. For this reason, our main quantitative comparisons focus on training-free methods such as MaskCut, DiffSeg, and DiffCut.
>
> However, to provide the context requested, we have added a comprehensive table in the revision (see below) comparing Falcon against both training-free and training-based methods.
>
> | Method | AM | AT |  VOC | Context | COCO-Object | COCO-Stuff-27 | Cityscapes | ADE20K |
> | :--- | :---: | :---: | :---: | :---: | :---: | :---: | :---: | :---: |
> | SAM3 (Category prompt) | ✓ |  ✓ | - | 60.80 | - | - | 65.20 | 39.00 |
> |  |  |  |  |  |  |  |  |  |
> | U2Seg | ✗ |  ✓ | - | - | - | 30.20 | - | - |
> | STEGO | ✗ |  ✓ | - | - | - | 28.20 | 21.00 | - |
> |  |  |  |  |  |  |  |  |  |
> | MaskCLIP | ✗ |  ✗ | 38.80 | 23.60 | 20.60 | 19.60 | 10.00 | 9.80 |
> | MaskCut (reported by DiffCut) | ✗ | ✗ | 53.80 | 43.40 | 30.10 | 41.70 | 18.70 | 35.70 |
> | DiffSeg | ✗ | ✗ | 49.80 | 48.80 | 23.20 | 44.20 | 16.80 | 37.70 |
> | DiffCut | ✗ | ✗ | 65.20 | 56.50 | 34.10 | 49.10 | 30.60 | 44.30 |
> |  |  |  |  |  |  |  |  |  |
> | AutoSC† (ours, refined) | ✗ | ✗ | 77.57 | 57.27 | 61.56 | 49.39 | 25.72 | 40.10 |
> | DiffCut† (ours, refined) | ✗ | ✗ | 71.68 | **58.17** | 61.65 | 49.18 | 30.77 | 44.40 |
> |  |  |  |  |  |  |  |  |  |
> | **Falcon (ours)** | ✗ | ✗ | **78.40** | 57.15 | **61.80** | **50.37** | **33.69** | **45.17** |
> |  |  |  |  |  |  |  |  |  |
>
> In the revision, we will list these training-required methods alongside training-free methods and annotate each method with whether it uses annotated masks or adaptive training.
>
> ### How is MaskCut adapted for unsupervised semantic segmentation?
> First of all, we follow the exact protocol established by DiffSeg and DiffCut. We clarify the following:
>
> #### **MaskCut results in the original submission.**
> In the Tab.1 of the submission, the row MaskCut is reported by DiffCut paper, following their own protocol. We used these numbers to preserve comparability with prior work. The DiffCut authors have access to implementation details that are not fully documented in the paper and code.
>
> #### **How MaskCut method becomes a semantic segmentation method?**
> For any graph-cut method, the core output is a pixel-wise cluster index map. The semantic segmentation metrics are then computed using the standard unsupervised evaluation protocol introduced in DiffSeg and adopted by series of works including DiffCut, which Falcon fully integrates.
>
> In detail, the output of any unsupervised segmenter (including MaskCut, DiffSeg, DiffCut, and Falcon) is treated as a cluster index map: every pixel is assigned to one of these clusters. Over the whole evaluation set, the protocol counts how often each predicted cluster overlaps with each ground-truth class, and then apply the Hungarian algorithm to find a one-to-one matching between clusters and semantic classes that maximizes this agreement. Pixel accuracy and mIoU are computed after this matching. For datasets with an explicit background class, the protocol only matches clusters to foreground classes; any clusters that are not matched are merged into a single background label.
>
> In our experiments, MaskCut is used in exactly the same way as DiffSeg, DiffCut, and Falcon: we run MaskCut on ViT tokens to obtain partitions of the image, upsample this cluster index map to the image resolution, and then feed it into the same Hungarian-matching evaluation pipeline. Thus, we can use the same evaluation protocol as DiffSeg and DiffCut.

---

> ### Author Response · Authors · 2025-11-27
> **Response to Reviewer uM5A (Part II)**
>
> ### The original MaskCut use DINOv1 and CRF. Reported results should use same backbone and CRF.
>
> Here we want to clarify that MaskCut results in Tab.1 of the submission are reported by DiffCut paper. We cannot determine what backbone and refinement method applied in the MaskCut results. To address the fairness concern, we re-implemented MaskCut experiments within our codebase, and evaluate them under exactly the same conditions as Falcon. In particular:
> - We run MaskCut with DINOv3-B encoder. Note original MaskCut uses ViT Key embeddings. We report results for both Key features and Token features.
> - We follow DiffCut and set the MaskCut recursion to $k=5$, which is the best setting reported for MaskCut in DiffCut paper.
> - We also implement CRF post-processing in our codebase and report Falcon with CRF. In our experiments, DenseCRF brings only marginal or negligible changes compared to no post-processing for Faclon. We will include these results in the revised version.
>
>
> | Method                               | Encoder        | VOC    | Context | COCO-Object | COCO-Stuff-27 | Cityscapes | ADE20K |
> |--------------------------------------|---------------|--------|---------|-------------|---------------|-----------|--------|
> | MaskCut k=5 (Reported by DiffCut)    | Unknown        | 0.5380 | 0.4340  | 0.3010      | 0.4170        | 0.1870    | 0.3570 |
> |  |  |  |  |  |  |  |  |
> | MaskCut k=5 (Our reported)           | DINOv3b-Key   | **0.8181** | 0.3046  | 0.5197      | 0.2170        | 0.0274    | 0.1483 |
> | MaskCut k=5 (Our reported)           | DINOv3b-Token | 0.7367 | 0.3326  | 0.5287      | 0.2554        | 0.0606    | 0.1748 |
> |  |  |  |  |  |  |  |  |
> | Falcon+CRF                           | DINOv3b-Token | 0.7515 | **0.4454**  | **0.6206**      | **0.3071**        | **0.2540**    | **0.4204** |
> |  |  |  |  |  |  |  |  |
>
> As shown, even when using the same encoder (DINOv3-B tokens) and adding the same CRF post-processing, Falcon clearly outperforms our re-implementation of MaskCut across benchmarks, confirming that our gains are not due to backbone or refiner differences.
>
>
>
> ## *W3: Additional MaskCut and DenseCRF runtime comparison and refinement cost.*
>
> Thanks for raising the point about runtime analysis. Our efficiency claim is primarily about the **mask-generation solver (graph cut)** itself, while the choice of refinement module is independent.
>
> ### **Clarify the role of PAMR/NAMR**
> **We do *not* present PAMR or NAMR as contributions of the submission.** PAMR is an existing refiner and already used in DiffCut. We include it mainly to align with prior SOTA for a fair comparison. NAMR is a simple non-linear variant that we use to show that Falcon does not rely on a specific refinement module. Section 3.4 is intended to describe a complete pipeline, not to claim credit for the refiners. Falcon can be paired with PAMR, NAMR, CRF, or even no refinement, and our conclusions about Falcon's efficiency and performance are refiner-agnostic.
>
> ### **Mask generation vs. refinement times**
> In the original Table 3, the column *"Mask generation"* explicitly reports **only the graph-cut time** (NCut solver) and does not include any refinement cost. This is because, across all our experiments, the runtime of PAMR/NAMR is very small compared to the solver: on average, the refinement step accounts for only **3.44%** of the graph-cut time. Thus, the total inference time is dominated by mask generation, which is exactly where Falcon reduces complexity. To make this decomposition completely explicit, we will added a new column of "Refinement Time" to Tab.3 in the revised revision.

---

> ### Author Response · Authors · 2025-11-27
> **Response to Reviewer uM5A (Part III)**
>
> ### **MaskCut runtime**
> Here we include MaskCut in the runtime analysis. We re-implemented MaskCut experiments in our codebase and measured its mask-generation time under exactly the same conditions as Falcon (same DINOv3-B encoder, same input 1024 resolution, same RTX4090 GPU). The results are shown in the following table.
>
> | Method              | VOC GraphCut (s) | VOC Refine (s) | Context GraphCut (s) | Context Refine (s) | COCO-Obj GraphCut (s) | COCO-Obj Refine (s) | COCO-Stuff GraphCut (s) | COCO-Stuff Refine (s) | Cityscapes GraphCut (s) | Cityscapes Refine (s) | ADE20K GraphCut (s) | ADE20K Refine (s) |
> |---------------------|-------------:|-----------:|-----------------:|---------------:|------------------:|----------------:|--------------------:|------------------:|--------------------:|------------------:|----------------:|--------------:|
> | MaskCut (with CRF)  | 720.75       | 80.18      | 2537.03          | 310.40         | 1078.50           | 125.21          | 1077.81             | 149.24            | 241.28              | 32.59             | 1001.57         | 120.20        |
> | Falcon (with PAMR)  | 137.00       | 4.37       | 542.83           | 15.77          | 231.39            | 6.61            | 230.57              | 7.18              | 52.49               | 1.70              | 212.83          | 6.57          |
> | Falcon (with NAMR)  | 136.77       | 12.88      | 541.50           | 44.65          | 231.56            | 21.05           | 231.20              | 23.40             | 52.39               | 5.02              | 211.34          | 20.55         |
> | Falcon (with CRF)   | 137.73       | 58.74      | 542.73           | 217.63         | 230.69            | 108.46          | 230.55              | 115.09            | 52.37               | 27.92             | 212.78          | 97.86         |
>
>
> As shown, Falcon achieves consistently lower inference time than MaskCut across all benchmarks.
>
>
>
> ### **Clarifying the statement about CRF being CPU-bound.**
> As the reviewer pointed out, our original remark that "CRF post processing is CPU-bound" was specifically referring to the official MaskCut implementation, where PyDenseCRF runs on CPU. We fully agree that more efficient DenseCRF implementations exist (e.g., the C++/CUDA accelerated version in Reviewer's [5]), and we will soften this statement in the revised version.
>
> In our own experiments, we integrate PyDenseCRF into our codebase and measured *Falcon with CRF* under the same hardware. As shown in the table above, CRF does add additional cost compared to PAMR/NAMR, and, importantly, **Falcon's advantage remains over MaskCut equipped with the same CRF refinement**. We will include these runtime results in Tab.3 in the revised appendix.
>
>
>
> ## *Q1: Complexity of the spectral thresholding procedure for choosing $K$.*
>
> In our Falcon implementation, the spectral step for choosing $K$ accounts for only about **5–6%** of the total runtime. For each image, we build the Laplacian only on $32\times 32$ nodes token grid, run a symmetric eigendecomposition once on GPU (`torch.linalg.eigh`), and then count how many eigenvalues lie below a cutoff $\kappa$. The thresholding itself is linear in the number of eigenvalues, i.e. $O(N)$. The dominant cost is the eigendecomposition, which is $O(N^3)$ in theory; however, $N$ is $1,024$ and this step runs on GPU, so in practice it is very small compared to the following proximal updates.

---

> ### Author Response · Authors · 2025-11-27
> **Response to Reviewer uM5A (Part IV)**
>
> ## *Q2: How the number of iterations T affects the performance? What is the iterations used in practice?*
>
> In Falcon, one outer iteration corresponds to a full application of the discrete proximal update in Eq. (11): computing NCut gradient scores for all tokens and projecting back to one-hot assignments. We run this outer loop until convergence or a small maximum number of iterations is reached.
>
> We evaluated how the number of outer iterations behaves in practice by sweeping the inertia initialization $\tau_0$ and backtracking factor $\gamma$ on Cityscapes with two encoders (DINOv3-B16 and SSD-1B). For each setting we recorded the average outer iterations per image and the resulting mIoU:
>
> | Encoder   | $\tau_0$ (inertia) | $\gamma$ (growth) | Outer iters / img | Backtrack rate | mIoU (%) |
> | :-------- | :------------------- | :------------------ | :---------------- | :------------- | :------- |
> | **DINOv3-B16** | 0.5                 | 1.5                | 1.22              | 43.3%          | 25.27    |
> |            | 0.5                 | 5.0                | 1.22              | 43.4%          | 25.27    |
> |            | **1.0**             | **2.0**            | **1.00**          | **0.0%**       | **25.29**|
> | **SSD-1B** | 0.5                 | 1.5                | 1.44              | 67.0%          | 30.56    |
> |            | 0.5                 | 5.0                | 1.21              | 67.8%          | 30.56    |
> |            | **1.0**             | **2.0**            | **1.02**          | **1.2%**       | **30.56**|
>
>
> According to the table above, we have the following observations:
> - Under our default setting ($\tau_0 = 1, \gamma = 2$), the solver converges in essentially one outer iteration per image (average $1.00$–$1.02$), so $T=1$ in practice.
> - The final mIoU is virtually identical across all tested settings, showing that performance is insensitive to $T$ once at least one iteration is performed. The variation is less than 0.1%.
>
> In all our reported experiments we set a small upper bound $T_{\max}=5$ and stop early when the objective converges. In practice the stopping condition is met after the first outer iteration.
>
>
>
> ## *Q3: Number of line-search steps per iteration.*
>
> The backtracking line search in Eq. (12) is a theoretical safety guard on the inertia weight $\tau_t$: if a tentative update does not decrease the NCut objective, we multiply $\tau_t$ by a factor $\gamma>1$ and retry. Otherwise, we accept the step immediately and perform **no** inner loop.
>
> As shown in the "Backtrack rate" column of the table above:
>
> - With our default parameters $\tau_0 = 1, \gamma = 2$, the backtrack rate is essentially 0% for both DINOv3-B and SSD-1B. This means the descent condition is satisfied on the first trial, so no extra line-search steps are performed in practice.
> - Even with an aggressively small initialization $\tau_0 = 0.5$, backtracking triggers more often (43–68% of outer iterations), but the average number of outer iterations per image only increases slightly (from about 1.0 to at most 1.4), and the final mIoU remains unchanged.
>
> Thus, the line search never behaves like a heavy inner loop: under standard settings it incurs essentially **zero** additional cost, and even in stress-test settings its overhead is negligible compared to computing one NCut gradient. We therefore treat $\tau_0$ and $\gamma$ as stability hyperparameters rather than sensitive knobs for runtime or accuracy. We will clarify these in the revised version.

---

### Official Review · Reviewer_zCyW · 2025-11-01

**Soundness:** 2
**Presentation:** 2
**Contribution:** 3
**Rating:** 6
**Confidence:** 5

**Summary:**

This paper introduces FALCON, a proximal-gradient solver for the discrete K-way normalized-cut (NCut) problem in unsupervised image segmentation. FALCON directly optimizes discrete assignments without spectral relaxation or eigen-decomposition. Experiments show the performance on six segmentation benchmarks (VOC, COCO-Object, Cityscapes, etc.)

**Strengths:**

1. The paper reformulates the classical NCut objective as a proximal-gradient problem with discrete feasibility preserved at every step, eliminating spectral relaxation and rounding. The idea is sound and interesting.
2. Experiments across six public benchmarks validate the performance of the proposed method.

**Weaknesses:**

1. It seems that the proposed method is mathematical. But the the intuition for why the proximal scheme works well is not clear.
2. The method relies on pretrained vision encoders (e.g., SSD-1B, DINOv3), raising questions about its standalone generality in low-resource or domain-shift settings.

**Questions:**

1. Sensitivity of hyperparameters. How sensitive is convergence to the choice of the backtracking parameter $\gamma$ and the inertia term?
2. Can FALCON handle dynamic or streaming data scenarios where affinities evolve online?
3. The number of segments $K$ is determined by counting eigenmodes below a cutoff $k$. How robust is this spectral thresholding across datasets, and could a data-adaptive or learnable k further improve stability?

---

> ### Author Response · Authors · 2025-11-25
> **Response to Reviewer zCyW (Part I)**
>
> We thank Reviewer zCyW for their constructive and positive feedback and for recognizing the soundness and novelty of our proximal-gradient formulation. We appreciate the opportunity to clarify the intuition, generality, and robustness of Falcon. We will selectively update the following discussions in the revised version.
>
>
> ## *W1: Intuition for why the proximal scheme works.*
> Classical NCut solvers typically rely on **a sequence of approximations** in order to use spectral methods: (i) they relax discrete labels into continuous eigenvectors, optimizing a surrogate rather than the original combinatorial objective; (ii) they then perform recursive bipartitioning, where each split is locally optimal for a subproblem but not globally optimal for the final $K$-way partition; and (iii) they finally round the continuous solution back to discrete labels, which can introduce additional quantization errors.
>
> In contrast, Falcon starts from an **exactly equivalent combinatorial reformulation** of NCut (Eq. 6) rather than a relaxation. The proximal-gradient scheme is applied directly to this discrete formulation: each update is derived from the true NCut energy, and a simple projection step maps the iterate back to a valid one-hot assignment. Instead of greedily splitting the graph into two parts multiple times, Falcon refines a global $K$-way partition over the whole graph.
>
> Intuitively, the proximal term acts as an anchor: it keeps each update close to the previous valid segmentation, stabilizing the optimization and turning the procedure into a principled local search within the original discrete feasible set. This way, every step stays faithful to the true NCut objective, reducing the approximation gap introduced by spectral relaxation, greedy recursion, and rounding.
>
> ## *W2: Generality in low-resource or domain-shift settings.*
>
> We position Falcon as a training-free solver designed for scenarios where a pretrained vision encoder is available. Encoders like DINOv3 are typically trained on raw images without mask annotations (often as open-weight models), and Falcon converts their embeddings into segmentation masks purely at inference time. This fits naturally into an unsupervised pseudo-labeling pipeline as in CutLER or UnSAM: representations are derived from unlabeled images, Falcon generates masks without human supervision, and these masks can serve as pseudo-labels for segmentation models. In low-resource settings, avoiding dense mask annotation is often more critical than the one-time cost of reusing a pretrained encoder.
>
> Regarding domain shift, Falcon largely inherits the robustness of the underlying encoder (as evidenced by consistent gains across encoders in Table 2), and it does not introduce additional task-specific overfitting because of its training-free nature. As shown in Fig. 5 in Appendix, Falcon still produces reasonable segmentations on medical CT scans, remote sensing imagery, and cartoons—domains that differ substantially from the natural images used for encoder pretraining. We will explicitly incorporate this discussion into the revised version to clarify Falcon's scope.

---

> ### Author Response · Authors · 2025-11-25
> **Response to Reviewer zCyW (Part II)**
>
> ## *Q1: Sensitivity of the backtracking parameter and the inertia term.*
> Falcon is highly robust to these hyperparameters. As defined in Eq. (11), the step size $\tau_t$ serves as the weight of the inertia term $\tau_t X^{(t)}$, anchoring the update to the previous assignment, while $\gamma$ acts as a backtracking growth factor that provides a safety guard to ensure monotone improvement of the objective. We use the same default values of $\tau_0=1$ and $\gamma=2$ across all datasets and pretrainings.
>
> To verify robustness, we performed a parameter sweep on Cityscapes using DINOv3-B16 and SSD-1B pretrainings, varying the initial inertia weight $\tau_0$ and growth factor $\gamma$:
>
> | Encoder   | $\tau_0$ (inertia) | $\gamma$ (growth) | Outer iters / img | Backtrack rate | mIoU (%) |
> | :-------- | :------------------- | :------------------ | :---------------- | :------------- | :------- |
> | **DINOv3-B16** | 0.5                 | 1.5                | 1.22              | 43.3%          | 25.27    |
> |            | 0.5                 | 5.0                | 1.22              | 43.4%          | 25.27    |
> |            | **1.0**             | **2.0**            | **1.00**          | **0.0%**       | **25.29**|
> | **SSD-1B** | 0.5                 | 1.5                | 1.44              | 67.0%          | 30.56    |
> |            | 0.5                 | 5.0                | 1.21              | 67.8%          | 30.56    |
> |            | **1.0**             | **2.0**            | **1.02**          | **1.2%**       | **30.56**|
>
> As shown in the table, the final segmentation quality is virtually identical across all settings: mIoU varies by less than 0.1\% for both pretrainings. Under standard settings ($\tau_0 \ge 1.0$), the algorithm typically converges in a single outer iteration with negligible or no backtracking, confirming the stability of our closed-form gradient score. Even with an aggressively small initialization such as $\tau_0 = 0.5$, the backtracking mechanism activates as intended (increasing the backtrack rate and slightly the iteration count) to restore stability without compromising accuracy. These parameters, therefore, act primarily as safety margins for the optimization trajectory rather than sensitive tuning knobs for performance.
>
>
>
> ## *Q2: Can Falcon handle dynamic or streaming data scenarios where affinities evolve online?*
>
> In its current form, Falcon is a static graph method. For each image, we construct the affinity matrix and run the discrete NCut optimization to convergence. While one can apply Falcon in a streaming setting by processing each frame independently (which is feasible given Falcon's high inference speed), we do not currently model the online evolution of the affinity graph. Extending the proximal scheme to dynamic graphs, e.g., by using previous solutions as warm starts or designing time-based update rules, is a promising direction. We will explicitly identify this as a future research direction in the revised version.

---

> ### Author Response · Authors · 2025-11-25
> **Response to Reviewer zCyW (Part III)**
>
> ## *Q3: Robustness of spectral thresholding for choosing $K$.*
>
> We determine $K$ using a standard spectral heuristic: we compute eigenvalues of the normalized affinity matrix, and count those below a cutoff $\kappa$ as the number of segments $K$. This step is computationally negligible compared to feature extraction and the following proximal iterations. As shown in Fig. 4 (right) in the submission, varying $\kappa$ within a broad, semantically reasonable interval directly changes the estimated $K$, but leaves the final segmentation quality (mIoU) relatively stable on both Pascal VOC and Cityscapes. Hence, these results indicate that using $\kappa$ within a reasonable range is robust in terms of performance.
>
> In practice, we fix a single $\kappa$ per dataset rather than tuning it per image. Empirically, we find a consistent pattern: datasets with sparse annotations in iconic scenes (a few large objects) such as Pascal VOC work well with a cutoff around $\kappa=0.0$, whereas more densely annotated, scene-centric datasets (Cityscapes, COCO-Stuff, COCO-Object, Context, ADE20K) with more small objects share a higher threshold around $\kappa=0.3$. This dataset-specific choice is natural, because each benchmark defines a different segmentation task and annotation protocol, so there is no single universally appropriate granularity across all datasets. As acknowledged in the segmentation literature, from classical NCut to modern models such as SAM that support multi-granularity masks, the correct number of segments is inherently ambiguous and depends on how fine or coarse the annotation standard is.
>
> We agree that more adaptive or learnable predictors for $K$ are promising. At the same time, as discussed above, the appropriate choice of $K$ is tightly coupled to the definition of the segmentation task and its annotation protocol, and is not determined by image statistics alone. For this reason, it is still useful to retain an explicit mechanism that controls segmentation granularity. In more modern settings, such as language-conditioned referring segmentation, the number of segments could be determined by the image and the text prompt together. We will update this to identify this as a future research direction.

---

### Official Review · Reviewer_v4WW · 2025-11-02

**Soundness:** 3
**Presentation:** 4
**Contribution:** 4
**Rating:** 8
**Confidence:** 4

**Summary:**

This paper presents a new pseudo-mask generation framework, Falcon, which incorporates fast proximal linearization and adaptive mask refinement to achieve better or the same unsupervised segmentation performance with much faster generation.

**Strengths:**

1. good motivation. Repeated NCut is slow. This paper presents a faster generation method with better performance.
2. clear formulation and presentation. This paper formulates the generation problem using many formulas and proofs.
3. efficiency analysis. Inference time of different methods are compared.
4. encoder ablation. The author presents ablation of different SSL encoder to validates the effectiveness of Falcon.

**Weaknesses:**

1. While Falcon is more efficient than recursive NCut methods, its complexity still scales quadratically with the number of tokens, making it less optimal for extremely large datasets or scenarios with very high token resolutions.

**Questions:**

None

---

> ### Author Response · Authors · 2025-11-25
> **Response to Reviewer v4WW**
>
> We sincerely thank the reviewer for the very positive and encouraging assessment of our work.
>
> Regarding the concern about $O(N^2)$ scaling, we acknowledge that dense token affinity inevitably incurs quadratic complexity. However, Falcon remains highly practical even for high-resolution inputs by avoiding the memory-intensive recursion. To demonstrate this, we measured the average peak GPU memory (GB) of different methods at various input resolutions on top of DINOv3-B, averaged over 6 datasets (VOC, COCO-Object, COCO-Stuff, Cityscapes, ADE20K, Context), with batch size 1 on the same GPU (an NVIDIA RTX 6000 Ada):
>
> | Resolution | MaskCut            | DiffCut            | **Falcon (ours)**   |
> |--------------------------:|--------------------|--------------------|---------------------|
> | $256^2$                   | **0.5112 ± 0.0000**    | **0.5112 ± 0.0000**    | **0.5112 ± 0.0000** |
> | $512^2$                   | **0.5362 ± 0.0018**    | 0.5374 ± 0.0018    | **0.5362 ± 0.0018** |
> | $768^2$                   | **0.5927 ± 0.0000**    | 0.8201 ± 0.0350    | **0.5927 ± 0.0000** |
> | $1024^2$                  | 1.0013 ± 0.0000    | 1.9510 ± 0.2622    | **0.9269 ± 0.0000** |
> | $2048^2$                  | 10.5234 ± 0.0000   | 23.7140 ± 3.5698   | **9.4759 ± 0.0000** |
>
> As shown, Falcon scales much more gracefully than DiffCut. Notably, at $2048 \times 2048$, Falcon is 2.5x more memory-efficient than DiffCut (9.48GB vs. 23.71GB) and even slightly more efficient than the simpler MaskCut baseline. This confirms that Falcon is well-suited for high-resolution scenarios compared to the previous methods.

---

### Author Response · Authors · 2025-12-03
**Rebuttal Summary for the AC**

Dear AC,

We sincerely thank you and the reviewers for the careful reading and detailed feedback on our paper. The questions and suggestions have been invaluable in helping us sharpen and improve the work.

### Submission Snapshot

Our submission proposes Falcon, a proximal-gradient solver that directly optimizes the discrete multi-way token-based NCut objective for **training-free unsupervised segmentation** from pretrained ViT encoder. Falcon avoids spectral relaxation, recursive bipartitioning, and ad-hoc rounding, and instead maintains one-hot feasibility at every step via a principled proximal update. Compared to prior training-free methods, Falcon achieves state-of-the-art performance on **17 of 18 benchmark–encoder pairs**, while being up to **2.5× more memory-efficient** and **5–10× faster** during inference.

The reviews consist of **one strong positive** (8, confidence 4) and **two positive reviews** (6, confidences 5 and 2) that acknowledge the novelty, soundness, and empirical strength of Falcon, plus **one negative review** (2, confidence 3) focused primarily on the completeness of comparisons and clarity of presentation rather than on the correctness of the method. Below, we summarize how our rebuttal and additional analyses address the main concerns.

### Response to Comprehensive Baseline Comparisons and Fairness
***(Addressing R-uM5A, R-mrYD)***

**Scope.** Falcon is explicitly scoped to the **training-free regime**, where a frozen encoder is used at inference time without additional training. Our main comparisons therefore focus on training-free baselines: MaskCut, DiffSeg, DiffCut, and AutoSC. In the rebuttal, to clarify Falcon’s position in the broader landscape, we added a consolidated table that includes training-based methods (e.g., SAM3, STEGO, U2Seg) and annotates which methods rely on annotated masks or adaptive training, showing that these operate in a different regime.

**Strict apples-to-apples comparison with MaskCut and CRF.** We integrated MaskCut in our codebase and evaluated it under exactly the same conditions as Falcon. Under these strictly matched settings, Falcon consistently achieves higher mIoU than MaskCut across benchmarks, confirming that the gains stem from the Falcon solver rather than from encoder or refinement differences.


### Response to Validation of Efficiency and Scalability
***(Addressing R-v4WW, R-uM5A, R-mrYD)***

To analyze memory scalability, we measured peak GPU memory (RTX 6000 Ada) for MaskCut, DiffCut, and Falcon with DINOv3-B at multiple resolutions across six datasets. At high resolution (2048×2048), Falcon is about 2.5× more memory-efficient than the previous SOTA DiffCut (9.48GB vs. 23.71GB), showing that Falcon is practical at high resolution.

For runtime, we first clarify that the reported times in the submission account only for graph cut (“mask generation”). To make the comparison clearer and fairer, we added MaskCut with CRF to the timing table and decomposed inference into graph cut and refinement. Falcon’s graph-cut step is consistently much faster than MaskCut’s recursive NCut across all benchmarks, and refinement time is almost small compared to this dominant cost.


### Response to Hyperparameter Robustness
***(Addressing R-zCyW, R-uM5A, R-mrYD)***

We evaluated sensitivity to the inertia weight and backtracking factor on Cityscapes with DINOv3-B and SSD-1B. Across a wide range of settings, mIoU varies by less than 0.1%. Under our default configuration, the solver converges in essentially one outer iteration per image with an almost zero backtrack rate, so line search has negligible overhead.

For selecting K, we use a simple spectral heuristic: a single GPU eigen-decomposition followed by counting eigenvalues below a cutoff $\kappa$. This step is lightweight (5–6% of total runtime) and, as shown in our experiments, mIoU remains stable over a broad range of $\kappa$ values. We use a single $\kappa$ per dataset (reflecting task differences in annotation granularity) and do not tune $\kappa$ per image, indicating that K selection is both robust and inexpensive.

### Response to Presentation Clarity
***(Addressing R-uM5A, R-mrYD)***

In the rebuttal, we clarified the meanings of key vectors and scalars and identified cases of symbol overuse, and we commit to cleaning up the notation. We will also provide a list of key hyperparameters and defaults. We will release the full codebase upon acceptance.

### Conclusion
Three reviewers (8, 6, 6) recognize the technical novelty and practical impact of Falcon and view it as above the acceptance threshold. The remaining negative review (2, confidence 3) is driven mainly by concerns about additional baseline completeness, which we have clarified where they fall outside our training-free scope and, within scope, addressed with new fair MaskCut+CRF baselines, and, memory and runtime analyses.

We sincerely appreciate your time and consideration.

---

### Meta-Review · Area_Chair_jFvA · 2026-01-05

**Summary:**

This paper presents a new pseudo-mask generation framework to incorporate fast proximal linearization and adaptive mask refinement to achieve better or the same unsupervised segmentation performance with much faster generation. This work has four reviewers with three positive reviewers (8, 6, 6) and a negative reviewer (2). The authors have provided rebuttals for comprehensive baseline comparisons, fairness, validation of the efficient and scalability, and presentation clarifying issues. I agree with three positive reviewers to accep this work.

**Reviewer Concerns:**

Most of concerns have been addressed by the rebuttal. The authors are suggested to prepare the final version based on the rebuttal.

**Reviewer Scores:**

This work has four reviewers with three positive reviewers (8, 6, 6) and a negative reviewer (2). No reviewer change the scores.

---

### Decision · Program_Chairs · 2026-01-26

Accept (Poster)